# Beyond Linearization: On Quadratic and Higher-Order Approximation of Wide Neural Networks

**Yu Bai**
Salesforce Research
yu.bai@salesforce.com

**Jason D. Lee**
Princeton University
jasonlee@princeton.edu

## Abstract

Recent theoretical work has established connections between over-parametrized neural networks and linearized models governed by the Neural Tangent Kernels (NTKs). NTK theory leads to concrete convergence and generalization results, yet the empirical performance of neural networks are observed to exceed their linearized models, suggesting insufficiency of this theory.

Towards closing this gap, we investigate the training of over-parametrized neural networks that are beyond the NTK regime yet still governed by the Taylor expansion of the network. We bring forward the idea of *randomizing* the neural networks, which allows them to escape their NTK and couple with quadratic models. We show that the optimization landscape of randomized two-layer networks are nice and amenable to escaping-saddle algorithms. We prove concrete generalization and expressivity results on these randomized networks, which lead to sample complexity bounds (of learning certain simple functions) that match the NTK and can in addition be better by a dimension factor when mild distributional assumptions are present. We demonstrate that our randomization technique can be generalized systematically beyond the quadratic case, by using it to find networks that are coupled with higher-order terms in their Taylor series.

## 1 Introduction

Deep Learning has made remarkable impact on a variety of artificial intelligence applications such as computer vision, reinforcement learning, and natural language processing. Though immensely successful, theoretical understanding of deep learning lags behind. It is not understood how non-linear neural networks can be efficiently trained to approximate complex decision boundaries with a relatively few number of training samples.

There has been a recent surge of research on connecting neural networks trained via gradient descent with the neural tangent kernel (NTK) (Jacot et al., 2018; Du et al., 2018a;b; Chizat & Bach, 2018b; Allen-Zhu et al., 2018a; Arora et al., 2019a;b). This line of analysis proceeds by coupling the training dynamics of the nonlinear network with the training dynamics of its linearization in a local neighborhood of the initialization, and then analyzing the expressiveness and generalization of the network via the corresponding properties of its linearized model.

Though powerful, NTK is not yet a completely satisfying theory for explaining the success of deep learning in practice. In theory, the expressive power of the linearized model is roughly the same as, and thus limited to, that of the corresponding random feature space (Allen-Zhu et al., 2018a; Wei et al., 2019) or the Reproducing Kernel Hilbert Space (RKHS) (Bietti & Mairal, 2019). While these spaces can approximate any regular (e.g. bounded Lipschitz) function up to arbitrary accuracy, the norm of the approximators can be exponentially large in the feature dimension for certain non-smooth but very simple functions such as a single ReLU (Yehudai & Shamir, 2019). Using NTK analyses, the sample complexity bound for learning these functions can be poor whereas experimental evidence suggests that the sample complexity is mild (Livni et al., 2014). In practice, kernel machines with the NTK have been experimentally demonstrated to yield competitive results on large-scale tasks such as image classification on CIFAR-10; yet there is still a non-neglible per-

formance gap between NTK and full training on the same convolutional architecture (Arora et al., 2019a; Lee et al., 2019). It is an increasingly compelling question whether we can establish theories for training neural networks beyond the NTK regime.

In this paper, we study the optimization and generalization of over-parametrized two-layer neural networks via relating to their *higher-order approximations*, a principled generalization of the NTK. Our theory starts from the fact that a two-layer neural network $f_{\mathbf{W}_0+\mathbf{W}}(\mathbf{x})$ (with smooth activation) can be Taylor expanded with respect to the weight matrix $\mathbf{W}$ as

$$f_{\mathbf{W}_0+\mathbf{W}}(\mathbf{x}) = \sum_{r=1}^{m} a_r \sigma((\mathbf{w}_{0,r} + \mathbf{w}_r)^\top \mathbf{x}) = \underbrace{\sum_{r=1}^{m} a_r \sigma(\mathbf{w}_{0,r}^\top \mathbf{x})}_{f_{\mathbf{W}_0}(\mathbf{x})} + \sum_{k=1}^{\infty} \underbrace{\sum_{r=1}^{m} a_r \frac{\sigma^{(k)}(\mathbf{w}_{0,r}^\top \mathbf{x})}{k!}(\mathbf{w}_r^\top \mathbf{x})^k}_{f_{\mathbf{W}_0,\mathbf{W}}^{(k)}(\mathbf{x})}.$$

Above, $f_{\mathbf{W}_0}$ does not depend on $\mathbf{W}$, and $f^{(1)}$ corresponds to the NTK model, which is the dominant $\mathbf{W}$-dependent term when $\{\mathbf{w}_r\}$ are small and leads to the coupling between the gradient dynamics for training neural net and its NTK $f^{(1)}$.

Our key observation is that the dominance of $f^{(1)}$ is deduced from comparing the *upper bounds*— rather than the *actual values*—of $f_{\mathbf{W}_0,\mathbf{W}}^{(k)}(\mathbf{x})$. It is a priori possible that there exists a subset of $\mathbf{W}$'s in which the dominating term is not $f^{(1)}$ but some other $f^{(k)}$, $k \geq 2$. If we were able to train in that set, the gradient dynamics would be coupled with the dynamics on $f^{(k)}$ rather than $f^{(1)}$ and thus could be very different. That learning is coupled with $f^{(k)}$ could further offer possibilities for expressing certain functions with parameters of lower complexities, or generalizing better, as $f^{(k)}$ is no longer a linearized model. In this paper, we build on this perspective and identify concrete regimes in which neural net learning is coupled with higher-order $f^{(k)}$'s rather than its linearization.

The contribution of this paper can be summarized as follows.

• We demonstrate that after *randomization*, the linear NTK $f^{(1)}$ is no longer the dominant term, and so the gradient dynamics of the neural net is no longer coupled with NTK. Through a simple sign randomization, the training loss of an over-parametrized two-layer neural network can be coupled with that of a quadratic model (Section 3). We prove that the randomized neural net loss exhibits a nice optimization landscape in that every second-order stationary point has training loss not much higher than the best quadratic model, making it amenable to efficient minimization (Section 4).

• We establish results on the generalization and expressive power of such randomized neural nets (Section 5). These results lead to sample complexity bounds for learning certain simple functions that matches the NTK without distributional assumptions and are advantageous when mild isotropic assumptions on the feature are present. In particular, using randomized networks, the sample complexity bound for learning polynomials (and their linear combination) on (relatively) uniform base distributions is $O(d)$ lower than using NTK.

• We show that the randomization technique can be generalized to find neural nets that are dominated by the $k$-th order term in their Taylor series ($k > 2$) which we term as *higher-order NTKs*. These models also have expressive power similar as the linear NTK, and potentially even better generalization and sample complexity (Section 6 & Appendix D).

## 1.1 PRIOR WORK

We review prior work on the optimization, generalization, and expressivity of neural networks.

**Neural Net and Kernel Methods** Neal (1996) first proposed the connection between infinite-width networks and kernel methods. Later work (Daniely et al., 2016; Williams, 1997; Lee et al., 2018; Novak et al., 2019; Matthews et al., 2018) extended this connection to various settings including deep networks and deep convolutional networks. These works established that gradient descent on only the output layer weights is well-approximated by a kernel method for large width.

More recently, several groups discovered the connection between gradient descent on all the parameters and the neural tangent kernel (Jacot et al., 2018). Li & Liang (2018); Du et al. (2018b) utilized the coupling of the gradient dynamics to prove that gradient descent finds global minimizers of the

training loss of two-layer networks, and Du et al. (2018a); Allen-Zhu et al. (2018b); Zou et al. (2018) generalized this to deep residual and convolutional networks. Using the NTK coupling, Arora et al. (2019b); Cao & Gu (2019a;b) proved generalization error bounds that match the kernel method.

Despite the close theoretical connection between NTK and training deep networks, Arora et al. (2019a); Lee et al. (2019); Chizat & Bach (2018b) empirically found a significant performance gap between NTK and actual training. This gap has been theoretically studied in Wei et al. (2019); Allen-Zhu & Li (2019); Yehudai & Shamir (2019); Ghorbani et al. (2019a) which established that NTK has provably higher generalization error than training the neural net for specific data distributions and architectures.

The idea of randomization is initiated by Allen-Zhu et al. (2018a), who use randomization to provably learn a three-layer network; however it is unclear how the sample complexity of their algorithm compares against the NTK. Inspired by their work, we study the potential gains of coupling with a non-linear approximation over the linear NTK — we compare the performance of a quadratic approximation model with the linear NTK on two-layer networks and find that under mild data assumptions the quadratic approximation reduces sample complexity under mild data assumptions.

**Outside the NTK Regime** It is believed that the success of SGD is largely due to its algorithmic regularization effects. A large body of work Li et al. (2017); Nacson et al. (2019); Gunasekar et al. (2018b;a; 2017); Woodworth et al. (2019) shows that asymptotically gradient descent converges to a max-margin solution with a strong regularization effect, unlike the NTK regularization[1].

For two-layer networks, a series of works used the mean field method to establish the evolution of the network parameters via a Wasserstein gradient flow (Mei et al., 2018b; Chizat & Bach, 2018a; Wei et al., 2018; Rotskoff & Vanden-Eijnden, 2018; Sirignano & Spiliopoulos, 2018). In the mean field regime, the parameters move significantly from their initialization, unlike NTK regime, however it is unclear if the dynamics converge to solutions of low training loss.

Finally, Li et al. (2019) showed how a combination of large learning rate and injected noise amplifies the regularization from the noise and outperforms the NTK of the corresponding architecture.

**Landscape Analysis** Many prior works have tried to establish favorable landscape properties such as every local minimum is a global minimum (Ge et al., 2017; Du & Lee, 2018; Soltanolkotabi et al., 2018; Hardt & Ma, 2016; Freeman & Bruna, 2016; Nguyen & Hein, 2017a;b; Haeffele & Vidal, 2015; Venturi et al., 2018). Combining with existing advances in gradient descent avoiding saddle-points (Ge et al., 2015; Lee et al., 2016; Jin et al., 2017), these show that gradient descent find the global minimum. Notably, Du & Lee (2018); Ge et al. (2017) show that gradient descent converges to solutions also of low test error, with lower sample complexity than their corresponding NTKs.

**Complexity Bounds** Recently, researchers have studied norm-based generalization based (Bartlett et al., 2017; Neyshabur et al., 2015; Golowich et al., 2017), tighter compression-based bounds (Arora et al., 2018), and PAC-Bayes bounds (Dziugaite & Roy, 2017; Neyshabur et al., 2017) that identify properties of the parameter that allow for efficient generalization.

## 2 PRELIMINARIES

**Problem setup** We consider the standard supervised learning task, in which we are given a labeled dataset $\mathcal{D} = \{(\mathbf{x}_1, y_1), \ldots, (\mathbf{x}_n, y_n)\}$, where $(x_i, y_i) \in \mathcal{X} \times \mathcal{Y}$ are sampled i.i.d. from some distribution $\mathbb{P}$, and we wish to find a predictor $f : \mathcal{X} \to \mathcal{Y}$. Without loss of generality, we assume that $\mathcal{X} = \mathbb{S}^{d-1}(B_x) \subset \mathbb{R}^d$ for some $B_x > 0$ (so that the features are $d$-dimensional with norm $B_x$.)

Let $\ell : \mathcal{Y} \times \mathbb{R} \to \mathbb{R}_{\geq 0}$ be a loss function such that $\ell(y, 0) \leq 1$, and $z \mapsto \ell(y, z)$ is convex, 1-Lipschitz, and three-times differentiable with the second and third derivatives bounded by one for

---

[1] As a concrete example, Woodworth et al. (2019) showed that for matrix completion the NTK solution estimates zero on all unobserved entries and the max-margin solution corresponds to the minimum nuclear norm solution.

all $y \in \mathcal{Y}$. This includes for example the logistic and soft hinge loss for classification. We let

$$L(f) := \mathbb{E}_{\mathcal{D}}[\ell(y, f(\mathbf{x}))] := \frac{1}{n} \sum_{i=1}^{n} \ell(y_i, f(\mathbf{x}_i)) \quad \text{and} \quad L_P(f) := \mathbb{E}_{(\mathbf{x}, y) \sim P}[\ell(y, f(\mathbf{x}))]$$

denote respectively the empirical risk and population risk for any predictor $f : \mathcal{X} \to \mathcal{Y}$.

**Over-parametrized two-layer neural network**  We consider learning an over-parametrized two-layer neural network of the form

$$f_{\mathbf{W}}(\mathbf{x}) = f_{\mathbf{a}, \mathbf{W}}(\mathbf{x}) := \frac{1}{\sqrt{m}} \mathbf{a}^{\top} \sigma(\mathbf{W}^{\top} \mathbf{x}) = \frac{1}{\sqrt{m}} \sum_{r=1}^{m} a_r \sigma(\mathbf{w}_r^{\top} \mathbf{x}), \tag{1}$$

where $\mathbf{W} = [\mathbf{w}_1, \dots, \mathbf{w}_r] \in \mathbb{R}^{d \times m}$ is the first layer and $\mathbf{a} = [a_1, \dots, a_m]^{\top} \in \mathbb{R}^m$ is the second layer. The $1/\sqrt{m}$ factor is chosen to account for the effect of over-parametrization and is consistent with the NTK-type scaling of (Du et al., 2018b; Arora et al., 2019b). In this paper we fix $\mathbf{a}$ and only train $\mathbf{W}$ (and thus use $f_{\mathbf{W}}$ to denote the network.)

Throughout this paper we assume that the activation is second-order smooth in the following sense.

**Assumption A** (Smooth activation). *The activation function $\sigma \in C^2(\mathbb{R})$, and there exists some absolute constant $C > 0$ such that $|\sigma'(t)| \le Ct^2$, $|\sigma''(t)| \le C|t|$, and $\sigma''(\cdot)$ is $C$-Lipschitz.*

An example is the cubic ReLU $\sigma(t) = \text{relu}^3(t) = \max\{t, 0\}^3$. The reason for requiring $\sigma$ to be higher-order smooth (and thus excluding ReLU) will be made clear in the subsequent text[2].

### 2.1 NOTATION

We typically reserve lowercases $a, b, \alpha, \beta, \dots$ for scalars, bold lowercases $\mathbf{a}, \mathbf{b}, \boldsymbol{\alpha}, \boldsymbol{\beta}, \dots$ for vectors, and bold uppercases $\mathbf{A}, \mathbf{B}, \dots$ for matrices. For a matrix $\mathbf{A} = [\mathbf{a}_1, \dots, \mathbf{a}_m] \in \mathbb{R}^{d \times m}$, its $2, p$ norm is defined as $\|\mathbf{A}\|_{2,p} := (\sum_{r=1}^{m} \|\mathbf{a}_r\|_2^p)^{1/p}$ for all $p \in [1, \infty]$. In particular we have $\|\cdot\|_{2,2} = \|\cdot\|_{\mathsf{Fr}}$. We let $\mathsf{B}_{2,p}(R) := \{\mathbf{W} : \|\mathbf{W}\|_{2,p} \le R\}$ denote a $2, p$-norm ball of radius $R$. We use standard Big-Oh notation: $a = O(b)$ for stating $a \le Cb$ for some absolute constant $C > 0$, and $a = \widetilde{O}(b)$ for $a \le Cb$ where $C$ depends at most logarithmically in $b$ and all other problem parameters. For a twice-differentiable function $f : \mathbb{R}^d \to \mathbb{R}$, $\mathbf{x}_\star$ is called a second-order stationary point if $\nabla f(\mathbf{x}_\star) = \mathbf{0}$ and $\nabla^2 f(\mathbf{x}_\star) \succeq \mathbf{0}$.

## 3 ESCAPING NTK VIA RANDOMIZATION

To motivate our study, we now briefly review the NTK theory for over-parametrized neural nets and provide insights on how to go beyond the NTK regime.

Let $\mathbf{W}_0$ denote the weights in a two-layer neural network at initialization and $\mathbf{W}$ denote its movement from $\mathbf{W}_0$ (so that the current weight matrix is $\mathbf{W}_0 + \mathbf{W}$.) The observation in NTK theory, or the theory of lazy training (Chizat & Bach, 2018b), is that for small $\mathbf{W}$ the neural network $f_{\mathbf{W}_0 + \mathbf{W}}$ can be Taylor expanded as

$$f_{\mathbf{W}_0 + \mathbf{W}}(\mathbf{x}) = \frac{1}{\sqrt{m}} \sum_{r \le m} a_r \sigma((\mathbf{w}_{0,r} + \mathbf{w}_r)^{\top} \mathbf{x})$$

$$= \underbrace{\frac{1}{\sqrt{m}} \sum_{r \le m} a_r \sigma(\mathbf{w}_{0,r}^{\top} \mathbf{x})}_{f_{\mathbf{W}_0}(\mathbf{x})} + \underbrace{\frac{1}{\sqrt{m}} \sum_{r \le m} a_r \sigma'(\mathbf{w}_{0,r}^{\top} \mathbf{x})(\mathbf{w}_r^{\top} \mathbf{x})}_{:= f_{\mathbf{W}}^L(\mathbf{x})} + O\left(\frac{1}{\sqrt{m}} \sum_{r \le m} (\mathbf{w}_r^{\top} \mathbf{x})^2\right),$$

---

[2]We note that the only restrictive requirement in Assumption A is the Lipschitzness of $\sigma''$, which guarantees second-order smoothness of the objectives. The bounds on derivatives (and specifically their bound near zero) are merely for technical convenience and can be weakened without hurting the results.

so that the network can be decomposed as the sum of the initial network $f_{\mathbf{W}_0}$, the linearized model $f_{\mathbf{W}}^L$, and higher order terms. Specifically (ignoring $f_{\mathbf{W}_0}$ for the moment), when $m$ is large and $\|\mathbf{w}_r\|_2 = O(m^{-1/2})$, we expect $f_{\mathbf{W}}^L = O(1)$ and higher order terms to be $o_m(1)$, which is indeed the regime when we train $f_{\mathbf{W}_0+\mathbf{W}}$ via gradient descent. Therefore, the trajectory of training $f_{\mathbf{W}_0+\mathbf{W}}$ is coupled with the trajectory of training $f_{\mathbf{W}_0} + f_{\mathbf{W}}^L$, which is a convex problem and enjoys convergence guarantees (Du et al., 2018b).

Our goal is to find subsets of $\mathbf{W}$ so that the dominating term is not $f^L$ but something else in the higher order part. The above expansion makes clear that this cannot be achieved through simple fixes such as tuning the leading scale $1/\sqrt{m}$ or the learning rate — the domination of $f^L$ appears to hold so long as the movements $\mathbf{w}_r$ are small.

**Randomized coupling with quadratic model** We now explain how the idea of *randomization*, initiated in (Allen-Zhu et al., 2018a), can help get rid of the domination of $f^L$. Let $\mathbf{W}$ be a fixed weight matrix. Suppose for each weight vector $\mathbf{w}_r$, we sample a random variable $\Sigma_{rr} \in \mathbb{R}$ and consider instead the random weight matrix

$$\mathbf{W\Sigma} := \mathbf{W}\mathrm{diag}(\{\Sigma_{rr}\}_{r=1}^m) = [\Sigma_{11}\mathbf{w}_1, \ldots, \Sigma_{rr}\mathbf{w}_r],$$

then the second-order Taylor expansion of $f_{\mathbf{W}_0+\mathbf{W\Sigma}}$ can be written as

$$f_{\mathbf{W}_0+\mathbf{W\Sigma}}(\mathbf{x}) = \frac{1}{\sqrt{m}}\sum_{r=1}^m a_r\sigma\big((\mathbf{w}_{0,r} + \mathbf{w}_r\Sigma_{rr})^\top\mathbf{x}\big)$$

$$= f_{\mathbf{W}_0}(\mathbf{x}) + \underbrace{\frac{1}{\sqrt{m}}\sum_{r=1}^m a_r\sigma'(\mathbf{w}_{0,r}^\top\mathbf{x})(\Sigma_{rr}\mathbf{w}_r^\top\mathbf{x})}_{=f_{\mathbf{W\Sigma}}^L(\mathbf{x})} + \underbrace{\frac{1}{2\sqrt{m}}\sum_{r=1}^m a_r\sigma''(\mathbf{w}_{0,r}^\top\mathbf{x})\Sigma_{rr}^2(\mathbf{w}_r^\top\mathbf{x})^2}_{:=f_{\mathbf{W\Sigma}}^Q(\mathbf{x})} + \ldots,$$

where we have defined in addition the quadratic part $f_{\mathbf{W\Sigma}}^Q$. Due to the existence of $\{\Sigma_{rr}\}$, each original weight $\mathbf{w}_r$ now has an additional a scalar that is different in $f^L$ and $f^Q$. Specifically, if we choose

$$\Sigma_{rr} \overset{\mathrm{iid}}{\sim} \mathrm{Unif}\{\pm 1\} \tag{2}$$

to be *random signs*, then we have $\Sigma_{rr}^2 \equiv 1$ and thus $f_{\mathbf{W\Sigma}}^Q(\mathbf{x}) \equiv f_{\mathbf{W}}^Q(\mathbf{x})$, whereas $\mathbb{E}[\Sigma_{rr}] = 0$ so that $\mathbb{E}[f_{\mathbf{W\Sigma}}^L(\mathbf{x})] \equiv 0$. Consequently, $f^Q$ is not affected by such randomization whereas $f_{\mathbf{W\Sigma}}^L$ is now mean zero and thus can have substantially lower magnitude than $f_{\mathbf{W}}^L$.

More precisely, when $\|\mathbf{w}_r\|_2 \asymp m^{-1/4}$, the scalings of $f^L$ and $f^Q$ compare as follows:

- We have $\mathbb{E}_{\mathbf{\Sigma}}[f_{\mathbf{W\Sigma}}^L(\mathbf{x})] = 0$ and

$$\mathbb{E}_{\mathbf{\Sigma}}\left[(f_{\mathbf{W\Sigma}}^L(\mathbf{x}))^2\right] = \frac{1}{m}\sum_{r=1}^m a_r^2\sigma'(\mathbf{w}_{0,r}^\top\mathbf{x})^2(\mathbf{w}_r^\top\mathbf{x})^2 = O\left(\frac{1}{m}\sum_{r=1}^m \|\mathbf{w}_r\|_2^2\right) = O(m^{-1/2}),$$

so we expect $f_{\mathbf{W\Sigma}}^L(\mathbf{x}) = O(m^{-1/4})$ over a random draw of $\mathbf{\Sigma}$.
- The quadratic part scales as

$$f_{\mathbf{W\Sigma}}^Q(\mathbf{x}) = f_{\mathbf{W}}^Q(\mathbf{x}) = \frac{1}{2\sqrt{m}}\sum_{r=1}^m a_r\sigma''(\mathbf{w}_{0,r}^\top\mathbf{x})(\mathbf{w}_r^\top\mathbf{x})^2 = O\left(\frac{1}{\sqrt{m}}\sum_{r=1}^m \|\mathbf{w}_r\|_2^2\right) = O(1).$$

Therefore, at the random weight matrix $\mathbf{W\Sigma}$, $f^Q$ dominates $f^L$ and thus the network is coupled with its quadratic part rather than the linear NTK.

### 3.1 LEARNING RANDOMIZED NEURAL NETS

The randomization technique leads to the following recipe for learning $\mathbf{W}$: train $\mathbf{W}$ so that $\|\mathbf{w}_r\|_2 = O(m^{-1/4})$ and $\mathbf{W\Sigma}$ has in expectation low loss. We make this precise by formulating the problem as minimizing a *randomized neural net risk*.

**Randomized risk** Let $\widetilde{L} : \mathbb{R}^{d \times m} \to \mathbb{R}$ denote the vanilla empirical risk for learning $f_{\mathbf{W}}$:

$$\widetilde{L}(\mathbf{W}) = \mathbb{E}_{\mathcal{D}}\left[\ell(y, f_{\mathbf{W}_0 + \mathbf{w}}(\mathbf{x}))\right],$$

where we have reparametrized the weight matrix into $\mathbf{W}_0 + \mathbf{W}$ so that learning starts at $\mathbf{W} = \mathbf{0}$.

Following our randomization recipe, we now formulate our problem as minimizing the expected risk

$$L(\mathbf{W}) := \mathbb{E}_{\mathbf{\Sigma}}[\widetilde{L}(\mathbf{W}\mathbf{\Sigma})] = \mathbb{E}_{\mathbf{\Sigma}, \mathcal{D}}\left[\ell(y, f_{\mathbf{W}_0 + \mathbf{W}\mathbf{\Sigma}}(\mathbf{x}))\right],$$

where $\mathbf{\Sigma} \in \mathbb{R}^{m \times m}$ is a diagonal matrix with $\Sigma_{rr} \overset{\text{iid}}{\sim} \mathrm{Unif}\{\pm 1\}$. To encourage $\|\mathbf{w}_r\|_2 = O(m^{-1/4})$ and improve generalization, we consider a regularized version of $\widetilde{L}$ and $L$ with $\ell_{2,4}$ regularization:

$$\widetilde{L}_\lambda(\mathbf{W}) := \widetilde{L}(\mathbf{W}) + \lambda \|\mathbf{W}\|_{2,4}^8 \quad \text{and} \quad L_\lambda(\mathbf{W}) = L(\mathbf{W}) + \lambda \|\mathbf{W}\|_{2,4}^8 = \mathbb{E}_{\mathbf{\Sigma}}[\widetilde{L}_\lambda(\mathbf{W}\mathbf{\Sigma})].$$

Our regularizer penalizes $\mathbf{W}$, i.e. the distance from initialization, similar as in (Hu et al., 2019).[3]

**Symmetric initialization** We initialize the parameters $(\mathbf{a}, \mathbf{W}_0)$ randomly in the following way: set

$$a_1 = \cdots = a_{m/2} = +1, \;\; a_{m/2+1} = \cdots = a_m = -1,$$
$$\mathbf{w}_{0,r} = \mathbf{w}_{0,r+m/2} \overset{\text{iid}}{\sim} \mathsf{N}\!\left(0, B_x^{-2} I_d\right), \; \forall r \in [m/2]. \tag{3}$$

Above, we set half of the $a_i$'s as $+1$ and half as $-1$, and the weights $\mathbf{w}_{0,r}$ are i.i.d. in the $+1$ half and copied exactly into the $-1$ half. Such an initialization is almost equivalent to i.i.d. random $\mathbf{W}_0$, but has the additional benefit that $f_{\mathbf{W}_0}(\mathbf{x}) \equiv 0$ and also leads to simple expressivity arguments. Our initialization scale $B_x^{-2}$ is chosen so that for a random draw of $\mathbf{w}_0$, we have $\mathbf{w}_0^\top \mathbf{x} \sim \mathsf{N}(0, 1)$, which is on average $O(1)$[4]. For technical convenience, we also assume henceforth that the realized $\{\mathbf{w}_{0,r}\}$ satisfies the bound

$$\max_{r \in [m]}(B_x \|\mathbf{w}_{0,r}\|_2) = O\!\left(\sqrt{d + \log(m/\delta)}\right) = \widetilde{O}(\sqrt{d}). \tag{4}$$

This happens with probability at least $1 - \delta$ under random initialization (see proof in Appendix A.3), and ensures that $\max_{r \in [m]} |\mathbf{w}_{0,r}^\top \mathbf{x}| \le \widetilde{O}(\sqrt{d})$ simultaneously for all $\mathbf{x}$.

## 4 OPTIMIZATION

In this section, we show that $L_\lambda$ enjoys a nice optimization landscape.

### 4.1 NICE LANDSCAPE OF CLEAN RISK

As the randomized loss $L$ induces coupling of the neural net $f_{\mathbf{W}_0 + \mathbf{W}\mathbf{\Sigma}}$ with the quadratic model $f_{\mathbf{W}}^Q$, we expect its behavior to resemble the behavior of gradient descent on the following *clean risk*:

$$L^Q(\mathbf{W}) := \frac{1}{n} \sum_{i=1}^n \ell(y_i, f_{\mathbf{W}}^Q(\mathbf{x}_i)) = \frac{1}{n} \sum_{i=1}^n \ell\!\left(y_i, \frac{1}{2\sqrt{m}} \left\langle \mathbf{x}_i \mathbf{x}_i^\top, \mathbf{W}\mathbf{D}_i \mathbf{W}^\top \right\rangle\right).$$

Above, we have defined diagonal matrices $\mathbf{D}_i = \mathrm{diag}(\{a_r \sigma''(\mathbf{w}_{0,r}^\top \mathbf{x}_i)\}_{r \in [m]}) \in \mathbb{R}^{m \times m}$ which are not trained.

We now show that the clean risk $L^Q$, albeit non-convex, possesses a nice optimization landscape.

**Lemma 1** (Landscape of clean risk). *Suppose there exists $\mathbf{W}_\star \in \mathbb{R}^{d \times m}$ such that $L^Q(\mathbf{W}_\star) \le \mathsf{OPT}$. Let $\mathbf{\Sigma}' \in \mathbb{R}^{m \times m}$ be a diagonal matrix with $\Sigma'_{rr} \overset{\text{iid}}{\sim} \mathrm{Unif}\{\pm 1\}$, then we have*

$$\mathbb{E}_{\mathbf{\Sigma}'}\left[\nabla^2 L^Q(\mathbf{W})[\mathbf{W}_\star \mathbf{\Sigma}', \mathbf{W}_\star \mathbf{\Sigma}']\right]$$
$$\le \left\langle \nabla L^Q(\mathbf{W}), \mathbf{W} \right\rangle - 2(L^Q(\mathbf{W}) - \mathsf{OPT}) + \widetilde{O}\!\left(dB_x^4 \|\mathbf{W}\|_{2,4}^2 \|\mathbf{W}_\star\|_{2,4}^2 \, m^{-1}\right). \tag{5}$$

---

[3]Our specific choice of $\|\cdot\|_{2,4}$ norm is needed for measuring the average magnitude of $f_{\mathbf{W}}^Q$, whereas the high (8-th) power is not essential and can be replaced by any $(4 + \epsilon)$-th power without affecting the result.

[4]Our choice covers two commonly used scales in neural net analyses: $B_x = 1$, $\mathbf{w}_{0,r} \sim \mathsf{N}(0, I_d)$ in e.g. (Arora et al., 2019b; Allen-Zhu et al., 2018a); $B_x = \sqrt{d}$, $\mathbf{w}_{0,r} \sim \mathsf{N}(0, I_d/d)$ in e.g. (Ghorbani et al., 2019b).

This result implies that, for $\mathbf{W}$ in a certain ball and large $m$, every point of higher loss than $\mathbf{W}_\star$ will have either a first-order or a second-order descent direction. In other words, every approximate second-order stationary point of $L^Q$ is also an approximate global minimum. Our proof utilizes the fact that $L^Q$ is similar to the loss function in matrix sensing / learning quadratic neural networks, and builds on recent understandings that the landscapes of these problems are often nice (Soltanolkotabi et al., 2018; Du & Lee, 2018; Allen-Zhu et al., 2018a). The proof is deferred to Appendix B.1.

## 4.2 NICE LANDSCAPE OF RANDOMIZED NEURAL NET RISK

With the coupling between $f_{\mathbf{W}_0 + \mathbf{W}\mathbf{\Sigma}}(\mathbf{x})$ and $f_{\mathbf{W}}^Q(\mathbf{x})$ in hand, we expect the risk $L(\mathbf{W}) = \mathbb{E}[\widetilde{L}(\mathbf{W}\mathbf{\Sigma})]$ to enjoy similar guarantees as the clean risk does $L^Q(\mathbf{W})$ in Lemma 1. We make this precise in the following result.

**Theorem 2** (Landscape of $L$). *Suppose there exists $\mathbf{W}_\star \in \mathsf{B}_{2,4}(B_{w,\star})$ such that $L^Q(\mathbf{W}_\star) \leq \mathsf{OPT}$, and that*

$$m \geq O\big([B_x^{12}B_w^{12} + d^4 B_x^4 B_w^4 + d^2 B_x^{20} B_w^{20}]\epsilon^{-4} + d^5 B_x^8 B_w^8 \epsilon^{-2}\big). \tag{6}$$

*for some fixed $\epsilon \in (0, 1]$ and $B_w \geq B_{w,\star}$, then for all $\mathbf{W} \in \mathsf{B}_{2,4}(B_w)$, we have*

$$\mathbb{E}_{\mathbf{\Sigma}'}\big[\nabla^2 L(\mathbf{W})[\mathbf{W}_\star \mathbf{\Sigma}', \mathbf{W}_\star \mathbf{\Sigma}']\big] \leq \langle \nabla L(\mathbf{W}), \mathbf{W} \rangle - 2(L(\mathbf{W}) - \mathsf{OPT}) + \epsilon. \tag{7}$$

As an immediate corollary, we have a similar characterization of the regularized loss $L_\lambda$.

**Corollary 3** (Landscape of $L_\lambda$). *For any $B_w \geq B_{w,\star}$, under the conditions of Theorem 2, we have for all $\lambda > 0$ and all $\mathbf{W} \in \mathsf{B}_{2,4}(B_w)$ that*

$$\begin{aligned}
&\mathbb{E}_{\mathbf{\Sigma}'}\big[\nabla^2 L_\lambda(\mathbf{W})[\mathbf{W}_\star \mathbf{\Sigma}', \mathbf{W}_\star \mathbf{\Sigma}']\big] \\
&\leq \langle \nabla L_\lambda(\mathbf{W}), \mathbf{W} \rangle - 2(L_\lambda(\mathbf{W}) - \mathsf{OPT}) - \lambda \|\mathbf{W}\|_{2,4}^8 + C\lambda \|\mathbf{W}_\star\|_{2,4}^8 + \epsilon,
\end{aligned} \tag{8}$$

*where $C = O(1)$ is an absolute constant.*

Theorem 2 follows directly from Lemma 1 through the coupling between $L$ and $L^Q$ (as well as their gradients and Hessians). Corollary 3 then follows by controlling in addition the effect of the regularizer. The full proof of Theorem 2 and Corollary 3 are deferred to Appendices B.4 and B.5.

We now present our main optimization result, which follows directly from Corollary 3.

**Theorem 4** (Optimization of $L_\lambda$). *Suppose there exists $\mathbf{W}_\star$ such that*

$$L^Q(\mathbf{W}_\star) \leq \mathsf{OPT} \quad \text{and} \quad \|\mathbf{W}_\star\|_{2,4} \leq B_{w,\star} \tag{9}$$

*for some $\mathsf{OPT} > 0$. For any $\gamma = \Theta(1)$ and $\epsilon > 0$, we can choose $\lambda$ suitably and $m \geq \widetilde{O}(\mathrm{poly}(d, B_x B_{w,\star}, \epsilon^{-1}))$ such that the regularized loss $L_\lambda$ satisfies the following: any second order stationary point $\widehat{\mathbf{W}}$ has low loss and bounded norm:*

$$L_\lambda(\widehat{\mathbf{W}}) \leq (1 + \gamma)\mathsf{OPT} + \epsilon \quad \text{and} \quad \left\|\widehat{\mathbf{W}}\right\|_{2,4} \leq O(B_{w,\star}). \tag{10}$$

**Proof sketch.** The proof of Theorem 4 consists of two stages: first "localize" any second-order stationary point into a (potentially very big) norm ball using the $\|\cdot\|_{2,4}^8$ regularizer, then use Corollary 3 in this ball to further deduce that $L_\lambda$ is low and $\left\|\widehat{\mathbf{W}}\right\|_{2,4} \leq O(\|\mathbf{W}_\star\|_{2,4})$. The full proof is deferred to Appendix B.6.

**Efficient optimization & allowing large learning rate** Theorem 4 states that when the over-parametrization is enough, any second-order stationary point (SOSP) $\widehat{\mathbf{W}}$ of $L_\lambda$ has loss competitive with $\mathsf{OPT}$, the performance of best quadratic model. Consequently, algorithms that are able to find SOSPs (escape saddles) such as noisy SGD (Jin et al., 2019) can efficiently minimize $L_\lambda$ to up to a multiplicative / additive factor of $\mathsf{OPT}$. Further, by sampling a fresh $\mathbf{\Sigma}$ at each iteration and using the stochastic gradient $\nabla_{\mathbf{W}}\widetilde{L}_\lambda(\mathbf{W}\mathbf{\Sigma})$ (rather than computing the full $\nabla L_\lambda(\mathbf{W})$), the noisy SGD iterates can be computed efficiently. We note in passing that our coupling results work in any $\ell_{2,4}$ ball of $O(1)$ size further allows the use of a *large learning rate*: as soon as the learning rate is bounded by $O(m^{1/4})$, we would have $\|\mathbf{W}_t\|_{2,4} \leq O(1)$ in a constant number of iterations, and thus our coupling and landscape results would hold. This is in contrast with the NTK regime which requires the learning rate to be bounded by $O(1)$ (Du et al., 2018b).

## 5   GENERALIZATION AND EXPRESSIVITY

We now shift attention to studying the generalization and expressivity of the (randomized) neural net $\widehat{\mathbf{W}}$ learned in Theorem 4.

### 5.1   GENERALIZATION

As $\widehat{\mathbf{W}}$ is always coupled (through randomization) with the quadratic model $f_{\widehat{\mathbf{W}}}^Q$, we begin by studying the generalization of the quadratic model.

**Generalization of quadratic models**   Let

$$\mathcal{F}^Q(B_w) := \left\{ \mathbf{x} \mapsto f_{\mathbf{W}}^Q(\mathbf{x}) : \|\mathbf{W}\|_{2,4} \le B_w \right\}$$

denote the class of quadratic models for $\mathbf{W}$ in a $\ell_{2,4}$ ball. We first present a lemma that relates the Rademacher complexity of $\mathcal{F}^Q(B_w)$ to the expected operator norm of certain feature maps.

**Lemma 5** (Bounding generalization of $f^Q$ via feature operator norm). *For any non-negative loss $\ell$ such that $z \mapsto \ell(y, z)$ is 1-Lipschitz and $\ell(y, 0) \le 1$ for all $y \in \mathcal{Y}$, we have the Rademacher complexity bound*

$$\mathbb{E}_{\boldsymbol{\sigma},\mathbf{x}}\left[\sup_{\|\mathbf{W}\|_{2,4} \le B_w} \frac{1}{n} \sum_{i=1}^n \sigma_i \ell(y_i, f_{\mathbf{W}}^Q(\mathbf{x}_i))\right] \le B_w^2 \mathbb{E}_{\boldsymbol{\sigma},\mathbf{x}}\left[\max_{r \in [m]} \left\| \frac{1}{n} \sum_{i=1}^n \sigma_i \sigma''(\mathbf{w}_{0,r}^\top \mathbf{x}_i) \mathbf{x}_i \mathbf{x}_i^\top \right\|_{\mathrm{op}}\right] + \frac{1}{\sqrt{n}},$$

*where $\sigma_i \overset{\mathrm{iid}}{\sim} \mathrm{Unif}\{\pm 1\}$ are Rademacher variables.*

**Operator norm based generalization**   Lemma 5 suggests a possibility for the quadratic model to generalize better than the NTK model: the Rademacher complexity of $\mathcal{F}^Q(B_w)$ depends on the "feature maps" $\frac{1}{n} \sum_{i=1}^n \sigma_i \sigma''(\mathbf{w}_{0,r}^\top \mathbf{x}_i) \mathbf{x}_i \mathbf{x}_i^\top$ through their matrix operator norm. Compared with the (naive) Frobenius norm based generalization bounds, the operator norm is never worse and can be better when additional structure on $\mathbf{x}$ is present. The proof of Lemma 5 is deferred to Appendix C.1.

We now state our main generalization bound on the (randomized) neural net loss $L$, which concretizes the above insight.

**Theorem 6** (Generalization of randomized neural net loss). *For any data-dependent $\widehat{\mathbf{W}}$ such that $\left\|\widehat{\mathbf{W}}\right\|_{2,4} \le B_w$, we have*

$$\mathbb{E}_{\mathbf{W}_0,\mathcal{D}}\left[L(\widehat{\mathbf{W}}) - L_P(\widehat{\mathbf{W}})\right] \le \widetilde{O}\left(\frac{B_x^2 B_w^2 M_{x,\mathrm{op}}}{\sqrt{n}} + \frac{1}{\sqrt{n}}\right) + \widetilde{O}\left(B_x^3 B_w^3 m^{-1/4} + d^2 B_x^2 B_w^2 m^{-1/2}\right),$$

*where $M_{x,\mathrm{op}} := \left(B_x^{-2}\mathbb{E}_{\mathbf{x}}\left[\left\|\frac{1}{n}\sum_{i=1}^n \mathbf{x}_i \mathbf{x}_i^\top\right\|_{\mathrm{op}}\right]\right)^{1/2}$ is the (rescaled) operator norm of the empirical covariance matrix. In particular, $M_{x,\mathrm{op}} \le 1$ always holds; if in addition $\mathbf{v}^\top \mathbf{x}$ is $K\sqrt{\mathrm{Var}(\mathbf{v}^\top \mathbf{x})}$ sub-Gaussian for all $\mathbf{v} \in \mathbb{S}^{d-1}(1)$ and $\kappa(\mathrm{Cov}(\mathbf{x})) \le \kappa$, then $M_{x,\mathrm{op}} \le \kappa/\sqrt{d}$ whenever $n \ge O(K^4 d)$.*

The generalization bound in Theorem 6 features two desirable properties:

(1) For large $m$ (e.g. $m \gtrsim n^4$), the bound scales at most logarithmically with the width $m$, therefore allowing learning with small samples and extreme over-parametrization;

(2) The main term $\widetilde{O}(B_x^2 B_w^2 M_{x,\mathrm{op}}/\sqrt{n})$ automatically adapts to properties of the feature distribution and can lower the generalization error than the naive bound by at most $O(1/\sqrt{d})$ without requiring us to tune any hyperparameter. Concretely, we have $M_{x,\mathrm{op}} \le O(1/\sqrt{d})$ when $\mathbf{x}$ has an isotropic distribution such as $\mathrm{Unif}(\mathbb{S}^{d-1}(B_x))$ or $\mathrm{Unif}\{\pm B_x/\sqrt{d}\}^d$.

Theorem 6 follows directly from Lemma 5 and a matrix concentration Lemma. The proof is deferred to Appendix C.2.

## 5.2 Expressivity and Sample Complexity through Quadratic Models

In order to concretize our generalization result, we now study the expressive power of quadratic models through the concrete example of learning functions of the form $\sum_{j \leq k} \alpha_j (\boldsymbol{\beta}_j^\top \mathbf{x})^{p_j}$, i.e. sum of "one-directional" polynomials (for consistency and comparability with (Arora et al., 2019b).)

**Theorem 7** (Expressing a sum of polynomials through $f^Q$). *Suppose* $\{(a_r, \mathbf{w}_{0,r})\}$ *are generated according to the symmetric initialization* (3) *and we use* $\sigma(t) = \frac{1}{6}\mathrm{relu}^3(t)$ *(so that* $\sigma''(t) = \mathrm{relu}(t)$.*) If* $f_\star(\mathbf{x}) = \sum_{j=1}^k \alpha_j (\boldsymbol{\beta}_j^\top \mathbf{x})^{p_j}$ *achieves training loss* $L(f_\star) \leq \epsilon_0$, *where* $p_j - 2 \in \{1\} \cup \{2\ell\}_{\ell \geq 0}$. *Then so long as the width is sufficiently large:*

$$m \geq \widetilde{O}\left( ndk^2 \sum_{j=1}^k p_j^3 \alpha_j^2 (B_x \|\boldsymbol{\beta}_j\|_2)^{2p_j} \epsilon^{-2} \right),$$

*we have with probability at least* $1 - \delta$ *(over* $\mathbf{W}_0$*) that there exists* $\mathbf{W}_\star \in \mathbb{R}^{d \times m}$ *such that*

$$L^Q(\mathbf{W}_\star) \leq \mathsf{OPT} := \epsilon_0 + \epsilon \quad \text{and} \quad \|\mathbf{W}_\star\|_{2,4}^4 \leq B_{w,\star}^4 = O\left( k \sum_{j=1}^k p_j^3 \alpha_j^2 B_x^{2(p_j-2)} \|\boldsymbol{\beta}\|_2^{2p_j} \delta^{-1} \right).$$

The proof of Theorem 7 is based on a reduction from expressing degree $p$ polynomials using quadratic models to expressing degree $p - 2$ polynomials using random feature models. The proof can be found in Appendix C.4.

**Comparison between quadratic and linearized (NTK) models**   We now illustrate our results in Theorem 6 and 7 in three concrete examples, in which we compare the sample complexity bounds of the randomized (quadratic) network and the linear NTK when $m$ is sufficiently large.

**Learning a single polynomial**. Suppose $f_\star(\mathbf{x}) = \alpha(\boldsymbol{\beta}^\top \mathbf{x})^p$ satisfies $L(f_\star) \leq \epsilon$, and we wish to find $\widehat{\mathbf{W}}$ with $O(\epsilon)$ test loss. By Theorem 7 we can choose $\mathbf{W}_\star$ such that $L^Q(\mathbf{W}_\star) \leq \mathsf{OPT} = 2\epsilon$, and by Theorem 4 we can find $\widehat{\mathbf{W}}$ such that $L(\widehat{\mathbf{W}}) \leq L_\lambda(\widehat{\mathbf{W}}) \leq 3\epsilon$ and $\|\widehat{\mathbf{W}}\|_{2,4} = O(B_{w,\star})$. Take $B_x = 1$, and assume $\mathbf{x}$ is sufficiently isotropic so that $M_{x,\mathrm{op}} = O(\frac{1}{\sqrt{d}})$, the sample complexity from Theorem 6 is

$$n \geq \widetilde{O}\left( \frac{B_x^4 B_w^4 M_{x,\mathrm{op}}^2}{\epsilon^2} \right) = \widetilde{O}\left( \frac{p^3 \alpha^2 \|\boldsymbol{\beta}\|_2^{2p}}{d\epsilon^2} \right) := n_Q.$$

In contrast, the sample complexity for linear NTK (Arora et al., 2019b; Cao & Gu, 2019a) to reach $\epsilon$ test loss is

$$n \geq \widetilde{O}\left( \frac{p^2 \alpha^2 \|\boldsymbol{\beta}\|_2^{2p}}{\epsilon^2} \right) := n_L.$$

We have $n_Q/n_L = \widetilde{O}(p/d)$, a reduction by a dimension factor unless $p \asymp d$. We note that the above comparison is simply comparing upper bounds, since in general the lower bound on the sample complexity of linear NTK is unknown.

**Learning a noisy 2-XOR**. Wei et al. (2019) established a sample complexity lower bound of linear NTK of $n \geq n_L = \Omega(d^2)$ to achieve constant generalization error on the noisy 2-XOR problem, which allows for a rigorous comparison against the quadratic model.

The ground truth function in 2-XOR is $f_\star(\mathbf{x}) = x_1 x_2 = ([(\mathbf{e}_1 + \mathbf{e}_2)^\top \mathbf{x}]^2 - [(\mathbf{e}_1 - \mathbf{e}_2)^\top \mathbf{x}]^2)/4$, where $\mathbf{x} \in \{\pm 1\}^d$, and $f_\star$ attains constant margin on the training distribution constructed in Wei et al. (2019). By Theorem 7, $f_\star$ can be $\epsilon$-approximated by $f_{\mathbf{W}_\star}^Q$ with $B_{w,\star}^4 \leq O(1)$. Thus by Theorem 6 the sample complexity for learning noisy 2-XOR through the randomized net $\widehat{\mathbf{W}}$ is

$$n \geq n_Q = \widetilde{O}\left( \frac{B_x^4 B_{w,\star}^4 M_{x,\mathrm{op}}^2}{\epsilon^2} \right) = \widetilde{O}\left( \frac{d}{\epsilon^2} \right).$$

This is $\widetilde{O}(d)$ better than the *sample complexity lower bound* of linear NTK and thus provably better.

**Low-rank matrix sensing**. Suppose we wish to learn a symmetric low-rank matrix $\mathbf{A}_\star \in \mathbb{R}^{d \times d}$ with $\|\mathbf{A}_\star\|_{\mathrm{op}} \leq 1$ and $\mathrm{rank}(\mathbf{A}_\star) \leq r$ through $n$ rank-one observations of the form $y_i = f_\star(\mathbf{x}_i) = \left\langle \mathbf{A}_\star, \mathbf{x}_i \mathbf{x}_i^\top \right\rangle$ where $\mathbf{x} \sim \mathrm{Unif}(\mathbb{S}^{d-1}(\sqrt{d}))$. This ground truth function can be written as $f_\star(\mathbf{x}_i) = \sum_{j=1}^r \alpha_j (\mathbf{v}_j^\top \mathbf{x}_i)^2$ where $|\alpha_j| \leq 1$ are the eigenvalues of $\mathbf{A}$ and $\mathbf{v}_j \in \mathbb{R}^d$ are the corresponding eigenvectors. For any 1-Lipschitz loss such as the absolute loss, by Theorem 7, there exists $\mathbf{W}_\star$ such that $L^Q(\mathbf{W}_\star) \leq \mathsf{OPT} = \epsilon$ and $B_{w,\star}^4 \leq O(r \sum_{j=1}^r \|\mathbf{v}_j\|_2^4) = O(r^2)$. Thus by Theorem 6, the sample complexity of reaching $2\epsilon$ test loss

$$\mathbb{E}_\mathbf{x}\big[\big|\big\langle \mathbf{A}_\star, \mathbf{x}\mathbf{x}^\top \big\rangle - f_{\widehat{\mathbf{W}}}(\mathbf{x})\big|\big] \leq 2\epsilon$$

through the randomized net $\widehat{\mathbf{W}}$ is

$$n \geq n_Q = \widetilde{O}\left( \frac{B_x^4 B_{w,\star}^4 M_{x,\mathrm{op}}^2}{\epsilon^2} \right) = \widetilde{O}\left( \frac{dr^2}{\epsilon^2} \right).$$

This compares favorably against the sample complexity upper bound for linear NTK, which needs

$$n \geq n_L = \widetilde{O}\left( \frac{B_x^4 \cdot (\sum_{j=1}^r \alpha_j \|\mathbf{v}_j\|_2^2)^2}{\epsilon^2} \right) = \widetilde{O}\left( \frac{d^2 r^2}{\epsilon^2} \right)$$

samples.

# 6 HIGHER-ORDER NTKS

We demonstrate that our idea of randomization for changing the dynamics of learning neural networks can be generalized systematically — through randomization we are able to obtain over-parametrized neural networks in which the $k$-th order term dominates the Taylor series. Due to space constraints, we provide our concrete results on the coupling, generalization, and expressivity of higher-order models in Appendix D.

# 7 CONCLUSION

In this paper we proposed and studied the optimization and generalization of over-parametrized neural networks through coupling with higher-order terms in their Taylor series. Through coupling with the quadratic model, we showed that the randomized two-layer neural net has a nice optimization landscape (every second-order stationary point has low loss) and is thus amenable to efficient minimization through escape-saddle style algorithms. These networks enjoy the same expressivity and generalization guarantees as linearized models but in addition can generalize better by a dimension factor when distributional assumptions are present. We extended the idea of randomization to show the existence of neural networks whose Taylor series is dominated by the $k$-th order term.

We believe our work brings in a number of open questions, such as how to better utilize the expressivity of quadratic models, or whether the study of higher-order expansions can lead to a more satisfying theory for explaining the success of full training. We also note that the Taylor series is only one avenue to obtaining accurate approximations of nonlinear neural networks. It would be of interest to design other approximation schemes for neural networks that are coupled with the network in larger regions of the parameter space.

ACKNOWLEDGMENT

The authors would like to thank Wei Hu, Tengyu Ma, Song Mei, and Andrea Montanari for their insightful comments. JDL acknowledges support of the ARO under MURI Award W911NF-11-1-0303, the Sloan Research Fellowship, and NSF CCF #1900145. The authors also thank the Simons Institute Summer 2019 program on the Foundations of Deep Learning, and the Institute of Advanced Studies Special Year on Optimization, Statistics, and Theoretical Machine Learning for hosting the authors.

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

## A    TECHNICAL TOOLS

### A.1    A MATRIX OPERATOR NORM CONCENTRATION BOUND

**Lemma 8** (Variant of Theroem 4.6.1, (Tropp et al., 2015)). *Suppose* $\{\mathbf{A}_{r,i}\}_{r\in[m],i\in[n]}$ *are fixed symmetric* $d \times d$ *matrices, and* $\{\sigma_i\}_{i\in[n]} \overset{\text{iid}}{\sim} \text{Unif}\{\pm 1\}$ *are Rademacher variables. Letting*

$$\mathbf{Y}_r = \sum_{i=1}^{n} \sigma_i \mathbf{A}_{r,i},$$

*then we have*

$$\mathbb{E}_{\boldsymbol{\sigma}}\left[\max_{r\in[m]} \|\mathbf{Y}_r\|_{\text{op}}\right] \le 4\sqrt{\max_{r\in[m]} v(\mathbf{Y}_r)\log(2md)},$$

*where*

$$v(\mathbf{Y}) := \left\|\mathbb{E}_{\boldsymbol{\sigma}}[\mathbf{Y}^2]\right\|_{\text{op}}.$$

*Proof.* Applying the high-probability bound in (Tropp et al., 2015, Theorem 4.6.1) and the union bound, we get

$$\mathbb{P}\left(\max_{r\in[m]} \|\mathbf{Y}_r\|_{\text{op}} \ge t\right) \le 2d \sum_{r\in[m]} \exp(-t^2/2v(\mathbf{Y}_r))$$

$$\le 2dm \exp(-t^2/2\max_{r\in[m]} v(\mathbf{Y}_r)) = \exp\left(-\frac{t^2}{2\max_{r\in[m]} v(\mathbf{Y}_r)} + \log(2dm)\right).$$

Let $V := \max_{r\in[m]} v(\mathbf{Y}_r)$, we have by integrating the above bound over $t$ that

$$\mathbb{E}\left[\max_{r\in[m]} \|\mathbf{Y}_r\|_{\text{op}}\right] \le \int_0^{\infty} \min\left\{\exp\left(-\frac{t^2}{2V} + \log(2dm)\right), 1\right\} dt$$

$$\le \sqrt{4V\log(2dm)} + \int_{\sqrt{4V\log(2dm)}}^{\infty} \exp(-t^2/2V + \log(2dm)) dt$$

$$\le \sqrt{4V\log(2dm)} + \int_{\sqrt{4V\log(2dm)}}^{\infty} \exp(-t^2/4V) dt$$

$$\le \sqrt{4V\log(2dm)} + \frac{\sqrt{4\pi V}}{2dm} \le 4\sqrt{V\log(2dm)}.$$

$\square$

## A.2 Expressing polynomials with random features

**Lemma 9.** *Let $\sigma(t) = \mathrm{relu}(t)$ and $\mathbf{w}_0 \sim \mathsf{N}(0, B_x^{-2} I_d)$ be Gaussian random features. For any $p \in \{1\} \cup \{2\ell\}_{\ell \geq 0}$ and $\boldsymbol{\beta} \in \mathbb{R}^d$, there exists a random variable $a = a(\mathbf{w}_0)$ such that*

$$\mathbb{E}_{\mathbf{w}_0}[\sigma(\mathbf{w}_0^\top \mathbf{x})a] = \alpha(\boldsymbol{\beta}^\top \mathbf{x})^p$$

*and $a$ satisfies the $\ell_2$ norm bound*

$$\mathbb{E}_{\mathbf{w}_0}[a^2] \leq 2\pi(p \vee 1)^3 \alpha^2 B_x^{2(p-1)} d \, \|\boldsymbol{\beta}\|_2^{2p} .$$

*Proof.* Consider the ReLU random feature kernel

$$K(\mathbf{x}, \mathbf{x}') = \mathbb{E}_{\mathbf{w}_0 \sim \mathsf{N}(0, B_x^{-2} I_d)}[\mathrm{relu}(\mathbf{w}_0^\top \mathbf{x})\mathrm{relu}(\mathbf{w}_0^\top \mathbf{x}')],$$

and let $\mathcal{H}_K$ denote the RKHS associated with this kernel. By the equivalence of feature maps (Minh et al., 2006, Proposition 1), for any feature map $\phi : \mathbb{S}^{d-1}(B_x) \mapsto \mathcal{H}$ (where $\mathcal{H}$ is a Hilbert space) that generates $K$ in the sense that

$$K(\mathbf{x}, \mathbf{x}') = \langle \phi(\mathbf{x}), \phi(\mathbf{x}') \rangle_{\mathcal{H}},$$

we have for any function $f$ that

$$\|f\|_{\mathcal{H}_K}^2 = \inf_{a \in \mathcal{H}} \left\{ \|a\|_{\mathcal{H}}^2 : f_\star(x) \equiv \langle a, \phi(\mathbf{x}) \rangle \right\}, \tag{11}$$

and the infimum over $a$ is attainable whenever it is finite.

For the ReLU random feature kernel $K$, let $u := \mathbf{x}^\top \mathbf{x}'/B_x^2$ and $\mathsf{N}_2(\rho)$ denote a bivariate normal distribution with marginals $\mathsf{N}(0, 1)$ and correlation $\rho \in [-1, 1]$. We have that

$$\begin{aligned}
K(\mathbf{x}, \mathbf{x}') &= \mathbb{E}_{\mathbf{w}_0 \sim \mathsf{N}(0, B_x^{-2} I_d)}\left[\mathrm{relu}(\mathbf{w}_0^\top \mathbf{x})\mathrm{relu}(\mathbf{w}_0^\top \mathbf{x}')\right] \\
&= \mathbb{E}_{(Z_1, Z_2) \sim \mathsf{N}_2(u)}[\mathrm{relu}(Z_1)\mathrm{relu}(Z_2)] \\
&= \frac{1}{2\pi}\left(u(\pi - \arccos u) + \sqrt{1 - u^2}\right) \\
&= \frac{1}{2\pi}\left(1 + \frac{\pi}{2}u + \sum_{\ell=1}^{\infty} \frac{(2\ell - 3)!!}{(2\ell - 2)!!(2\ell - 1)(2\ell)} u^{2\ell}\right) \\
&= \sum_{p \in \{0,1\} \cup \{2\ell\}_{\ell \geq 1}} c_p(\mathbf{x}^\top \mathbf{x}')^p B_x^{-2p} \\
&= \sum_{p \in \{0,1\} \cup \{2\ell\}_{\ell \geq 1}} \left\langle \sqrt{c_p} B_x^{-p} \mathbf{x}^{\otimes p}, \sqrt{c_p} B_x^{-p} (\mathbf{x}')^{\otimes p} \right\rangle,
\end{aligned}$$

where the constants $\{c_p\}$ satisfy

$$c_0 = 1/(2\pi), \quad c_1 = 1/4, \quad c_{2\ell} \geq \frac{1}{2\pi(2\ell - 1)^2(2\ell)} \quad \text{for } \ell \geq 1,$$

(and thus $c_p \geq (2\pi(p \vee 1)^3)^{-1}$ for all $p$), and $\mathbf{x}^{\otimes k} \in \mathbb{R}^{d^k}$ denote the $k$-wise tensor product of $\mathbf{x}$. Therefore, if we define feature map

$$\phi(\mathbf{x}) := \left[\sqrt{c_p} B_x^{-p} \mathbf{x}^{\otimes p}\right]_{p \in \{0,1\} \cup \{2\ell\}_{\ell \geq 1}},$$

we have $K(\mathbf{x}, \mathbf{x}') = \langle \phi(\mathbf{x}), \phi(\mathbf{x}') \rangle$. With this feature map, the function $f_\star(\mathbf{x}) = \alpha(\beta^\top \mathbf{x})^p$ can be represented as

$$f_\star(\mathbf{x}) \equiv \langle c_\star, \phi(\mathbf{x}) \rangle \quad \text{where} \quad c_\star = \left[0, \ldots, 0, \alpha \cdot c_p^{-1/2} B_x^p \boldsymbol{\beta}^{\otimes p}, 0, \ldots\right].$$

Thus by the feature map equivalence (11), we have $f_\star \in \mathcal{H}_K$ and

$$\|f\|_{\mathcal{H}_K}^2 \leq \|c_\star\|^2 = \alpha^2 c_p^{-1} B_x^{2p} \|\boldsymbol{\beta}\|_2^{2p} \leq 2\pi(p \vee 1)^3 \alpha^2 B_x^{2p} \|\boldsymbol{\beta}\|_2^{2p} .$$

Now apply the feature map equivalence (11) again with the random feature map

$$\mathbf{x} \mapsto \left\{\mathrm{relu}(\mathbf{w}_0^\top \mathbf{x})\right\}_{\mathbf{w}_0}$$

(which maps into the inner product space of $\mathbf{w}_0 \sim \mathsf{N}(0, B_x^{-2} I_d)$), we conclude that there exists $a = a(\mathbf{w}_0)$ such that $f_\star = \mathbb{E}_{\mathbf{w}_0}[\mathrm{relu}(\mathbf{w}_0^\top \mathbf{x})a]$ and

$$\mathbb{E}_{\mathbf{w}_0}[a^2] \leq \|f_\star\|_{\mathcal{H}_K}^2 \leq 2\pi(p \vee 1)^3 \alpha^2 B_x^{2p} \|\boldsymbol{\beta}\|_2^{2p} .$$

$\square$

### A.3 PROOF OF EQUATION (4)

Let $\mathcal{N}$ be an $1/2$-covering of $\mathbb{S}^{d-1}(1)$. We have $|\mathcal{N}| \leq 5^d$ and for any vector $\mathbf{w} \in \mathbb{R}^d$ that $\|\mathbf{w}\|_2 \leq 2 \sup_{\mathbf{v} \in \mathcal{N}} (\mathbf{v}^\top \mathbf{w})$ (see e.g. (Mei et al., 2018a, Section A).) We thus have

$$\mathbb{P}\left( \max_{r \in [m]} B_x \|\mathbf{w}_{0,r}\|_2 \geq t \right)$$

$$\leq \left( \max_{\mathbf{v} \in \mathcal{N}} B_x(\mathbf{v}^\top \mathbf{w}_{0,r}) \geq t/2 \right) \leq \exp(-t^2/8 + \log |\mathcal{N}| + \log m) \leq \exp(-t^2/8 + d \log 5 + \log m).$$

Setting $t = \sqrt{8(d \log 5 + \log(m/\delta))} = O(\sqrt{d + \log(m/\delta)})$ ensures that the above probability does not exceed $\delta$ as desired. $\qquad\square$

## B PROOFS FOR SECTION 4

### B.1 PROOF OF LEMMA 1

Computing the gradient of $L^Q$, we obtain

$$\nabla L^Q(\mathbf{W}) = \frac{2}{n} \sum_{i=1}^n \ell'(y_i, f_{\mathbf{W}}^Q(\mathbf{x}_i)) \frac{1}{2\sqrt{m}} \mathbf{x}_i \mathbf{x}_i^\top \mathbf{W} \mathbf{D}_i.$$

Further computing the Hessian gives

$$\nabla^2 L^Q(\mathbf{W})[\mathbf{W}_\star \mathbf{\Sigma}', \mathbf{W}_\star \mathbf{\Sigma}'] = \frac{2}{n} \sum_{i=1}^n \ell'(y_i, f_{\mathbf{W}}^Q(\mathbf{x}_i)) \cdot \underbrace{\frac{1}{2\sqrt{m}} \langle \mathbf{x}_i \mathbf{x}_i^\top, \mathbf{W}_\star \mathbf{\Sigma}' \mathbf{D}_i \mathbf{\Sigma}' \mathbf{W}_\star^\top \rangle}_{f_{\mathbf{W}_\star}^Q(\mathbf{x}_i)}$$

$$+ \frac{4}{n} \sum_{i=1}^n \ell''(y_i, f_{\mathbf{W}}^Q(\mathbf{x}_i)) \cdot \left( \underbrace{\frac{1}{2\sqrt{m}} \langle \mathbf{x}_i \mathbf{x}_i^\top, \mathbf{W} \mathbf{D}_i \mathbf{W}_\star^\top \mathbf{\Sigma}' \rangle}_{:= \widetilde{y}_i} \right)^2$$

$$= \underbrace{\frac{2}{n} \sum_{i=1}^n \ell'(y_i, f_{\mathbf{W}}^Q(\mathbf{x}_i)) f_{\mathbf{W}_\star}^Q(\mathbf{x}_i)}_{\mathrm{I}} + \underbrace{\frac{4}{n} \sum_{i=1}^n \ell''(y_i, f_{\mathbf{W}}^Q(\mathbf{x}_i)) \widetilde{y}_i^2}_{\mathrm{II}}.$$

Taking expectation over $\mathbf{\Sigma}'$, and using that $\ell'' \leq 1$, term II can be bounded as

$$\mathbb{E}_{\mathbf{\Sigma}'}[\mathrm{II}] \leq \mathbb{E}_{\mathbf{\Sigma}'} \left[ \frac{4}{n} \sum_{i=1}^n \widetilde{y}_i^2 \right]$$

$$= \mathbb{E}_{\mathbf{\Sigma}', \mathcal{D}} \left[ \frac{2}{m} \sum_{r \leq m} \sigma''(\mathbf{w}_{0,r}^\top \mathbf{x})^2 (\mathbf{w}_r^\top \mathbf{x})^2 (\Sigma'_{rr} \mathbf{w}_{\star,r}^\top \mathbf{x})^2 \right]$$

$$\leq C \cdot \mathbb{E}_{\mathcal{D}} \left[ \frac{1}{m} \sum_{r \leq m} (\mathbf{w}_{0,r}^\top \mathbf{x})^2 (\mathbf{w}_r^\top \mathbf{x})^2 (\mathbf{w}_{\star,r}^\top \mathbf{x})^2 \right]$$

$$\leq C B_x^4 \max_{r \in [m], i \in [n]} (\mathbf{w}_{0,r}^\top \mathbf{x}_i)^2 \cdot \frac{1}{m} \sum_{r \leq m} \|\mathbf{w}_r\|_2^2 \|\mathbf{w}_{\star,r}\|_2^2$$

$$\leq \widetilde{O}\left( d B_x^4 \|\mathbf{W}\|_{2,4}^2 \|\mathbf{W}_\star\|_{2,4}^2 m^{-1} \right),$$

where the last step used Cauchy-Schwarz on $\{\|\mathbf{w}_r\|_2\}$ and $\{\|\mathbf{w}_{\star,r}\|_2\}$.

Term I does not involve $\mathbf{\Sigma}'$ and can be deterministically bounded as

$$
\begin{aligned}
\mathrm{I} &= 2\mathbb{E}_{\mathcal{D}}[\ell'(y, f_{\mathbf{W}}^Q(\mathbf{x})) f_{\mathbf{W}_\star}^Q(\mathbf{x})] \\
&= 2\mathbb{E}_{\mathcal{D}}[\ell'(y, f_{\mathbf{W}}^Q(\mathbf{x})) f_{\mathbf{W}}^Q(\mathbf{x})] + 2\mathbb{E}_{\mathcal{D}}[\ell'(y, f_{\mathbf{W}}^Q(\mathbf{x}))(f_{\mathbf{W}_\star}^Q(\mathbf{x}) - f_{\mathbf{W}}^Q(\mathbf{x}))] \\
&\overset{(i)}{\leq} \langle \nabla L^Q(\mathbf{W}), \mathbf{W} \rangle + 2\mathbb{E}_{\mathcal{D}}[\ell(y, f_{\mathbf{W}_\star}^Q(\mathbf{x})) - \ell(y, f_{\mathbf{W}}^Q(\mathbf{x}))] \\
&\overset{(ii)}{=} \langle \nabla L^Q(\mathbf{W}), \mathbf{W} \rangle - 2(L^Q(\mathbf{W}) - \mathsf{OPT}).
\end{aligned}
$$

where (i) follows directly by computing $\langle \nabla L^Q(\mathbf{W}), \mathbf{W} \rangle$ and the convexity of $z \mapsto \ell(y, z)$, and (ii) follows from the assumption that $L^Q(\mathbf{W}_\star) \leq \mathsf{OPT}$. Combining the bounds for terms I and II gives the desired result. $\qquad\square$

### B.2 COUPLING LEMMAS

**Lemma 10** (Bound on $f^Q$). *For any $\mathbf{W} \in \mathbb{R}^{d \times m}$, the quadratic model $f_{\mathbf{W}}^Q$ satisfies the bound*

$$
\left| f_{\mathbf{W}}^Q(\mathbf{x}) \right| \leq \widetilde{O}\left( \sqrt{d} B_x^2 \|\mathbf{W}\|_{2,4}^2 \right)
$$

*for all $\mathbf{x} \in S^{d-1}(B_x)$.*

*Proof.* We have

$$
\begin{aligned}
\left| f_{\mathbf{W}}^Q(\mathbf{x}) \right| &= \left| \frac{1}{\sqrt{m}} \sum_{r \leq m} a_r \sigma''(\mathbf{w}_{0,r}^\top \mathbf{x})(\mathbf{w}_r^\top \mathbf{x})^2 \right| \\
&\leq \frac{1}{\sqrt{m}} \sum_{r \leq m} C |\mathbf{w}_{0,r}^\top \mathbf{x}| \cdot (\mathbf{w}_r^\top \mathbf{x})^2 \leq C\sqrt{m} B_x^2 \max_{r \in [m]} |\mathbf{w}_{0,r}^\top \mathbf{x}| \cdot \frac{1}{m} \sum_{r \leq m} \|\mathbf{w}_r\|_2^2 \\
&\leq C\sqrt{m} B_x^2 \widetilde{O}(\sqrt{d}) \cdot \left( \frac{1}{m} \sum_{r \leq m} \|\mathbf{w}_r\|_2^4 \right)^{1/2} = \widetilde{O}\left( \sqrt{d} B_x^2 R_{w,0} \|\mathbf{W}\|_{2,4}^2 \right).
\end{aligned}
$$

$\qquad\square$

**Lemma 11** (Coupling between $f$ and $f^Q$). *We have for all $\mathbf{x} \in \mathbb{S}^{d-1}(B_x)$ that*

(a) $\mathbb{E}_{\mathbf{\Sigma}}[f_{\mathbf{W}\mathbf{\Sigma}}^L(\mathbf{x})] = 0$ *and* $\mathbb{E}_{\mathbf{\Sigma}}[(f_{\mathbf{W}\mathbf{\Sigma}}^L(\mathbf{x}))^2] \leq \widetilde{O}(d^2 B_x^2 \|\mathbf{W}\|_{2,4}^2 m^{-1/2})$.

(b) $|\Delta_{\mathbf{W}\mathbf{\Sigma}}^Q(\mathbf{x})| \leq O(B_x^3 \|\mathbf{W}\|_{2,4}^3 m^{-1/4})$ *(almost surely for all $\mathbf{\Sigma}$.)*

*Proof.*    (a) Recall that

$$
f_{\mathbf{W}\mathbf{\Sigma}}^L(\mathbf{x}) = \frac{1}{\sqrt{m}} \sum_{r \leq m} a_r \sigma'(\mathbf{w}_{0,r}^\top \mathbf{x})(\mathbf{\Sigma}_{rr} \mathbf{w}_r^\top \mathbf{x}).
$$

As $\Sigma_{rr}$ has mean zero, we have $\mathbb{E}_{\mathbf{\Sigma}}[f^L] = 0$ and

$$
E_{\mathbf{\Sigma}}[(f^L)^2] = \frac{1}{m} \sum_{r \leq m} a_r^2 \sigma'(\mathbf{w}_{0,r}^\top \mathbf{x})^2 (\mathbf{w}_r^\top \mathbf{x})^2
$$

$$
\overset{(i)}{\leq} \frac{1}{m} \sum_r C(\mathbf{w}_{0,r}^\top \mathbf{x})^4 (\mathbf{w}_r^\top \mathbf{x})^2 \overset{(ii)}{\leq} C \max_{r \in [m]} (\mathbf{w}_{0,r}^\top \mathbf{x})^4 \cdot \frac{1}{m} \sum_r B_x^2 \|\mathbf{w}_r\|_2^2
$$

$$
\overset{(iii)}{\leq} \widetilde{O}(d^2 B_x^2) \cdot \frac{1}{m} \sum_{r \leq m} \|\mathbf{w}_r\|_2^2 \overset{(iv)}{\leq} C B_x^6 R_{w,0}^4 \cdot \left( \frac{1}{m} \sum_{r \leq m} \|\mathbf{w}_r\|_2^4 \right)^{1/2} = \widetilde{O}\left( d^2 B_x^2 \|\mathbf{W}\|_{2,4}^2 m^{-1/2} \right).
$$

Above, (i) follows from the assumption that $|\sigma'(t)| \leq Ct^2$, (ii) is Cauchy-Schwarz, (iii) uses the bound (4), and (iv) uses the power mean inequality on $\|\mathbf{w}_r\|_2$.

(b) We have by the Lipschitzness of $\sigma''$ that

$$
\begin{aligned}
|\Delta_{\mathbf{W}}^Q(\mathbf{x})| &= \Big| \frac{1}{\sqrt{m}} \sum_{r \le m} a_r \Big( \sigma((\mathbf{w}_{0,r} + \Sigma_{rr}\mathbf{w}_r)^\top \mathbf{x}) - \sigma(\mathbf{w}_{0,r}^\top \mathbf{x}) \\
&\qquad - \sigma'(\mathbf{w}_{0,r}^\top \mathbf{x})(\Sigma_{rr}\mathbf{w}_r^\top \mathbf{x}) - \sigma''(\mathbf{w}_{0,r}^\top \mathbf{x})(\Sigma_{rr}\mathbf{w}_r^\top \mathbf{x})^2 \Big) \Big| \\
&\le \frac{1}{\sqrt{m}} \sum_{r \le m} C |\Sigma_{rr}\mathbf{w}_r^\top \mathbf{x}|^3 \overset{(i)}{\le} C\sqrt{m}B_x^3 \cdot \frac{1}{m} \sum_{r \le m} \|\mathbf{w}_r\|_2^3 \\
&\le C\sqrt{m}B_x^3 \cdot \left( \frac{1}{m} \sum_{r \le m} \|\mathbf{w}_r\|_2^4 \right)^{3/4} \\
&= O\left( B_x^3 \|\mathbf{W}\|_{2,4}^3 \, m^{-1/4} \right),
\end{aligned}
$$

where again (i) uses the power mean inequality on $\|\mathbf{w}_r\|_2$.

$\square$

## B.3 CLOSENESS OF LANDSCAPES

**Lemma 12** ($L^Q$ close to $L$). *We have for all $\mathbf{W} \in \mathbb{R}^{d \times m}$ that*

$$
|L(\mathbf{W}) - L^Q(\mathbf{W})| \le \widetilde{O}\Big( B_x^3 \|\mathbf{W}\|_{2,4}^3 \, m^{-1/4} + d^2 B_x^2 \|\mathbf{W}\|_{2,4}^2 \, m^{-1/2} \Big).
$$

*Proof.* Recall that

$$
L(\mathbf{W}) = \mathbb{E}_{\Sigma,\mathcal{D}}[\ell(y, f_{\mathbf{W}_0 + \mathbf{W}\Sigma}(\mathbf{x}))] \quad \text{and} \quad L^Q(\mathbf{W}) = \mathbb{E}_{\mathcal{D}}[\ell(y, f_{\mathbf{W}}^Q(\mathbf{x}))].
$$

By the 1-Lipschitzness of $z \mapsto \ell(y, z)$ we have

$$
\begin{aligned}
|L(\mathbf{W}) - L^Q(\mathbf{W})| &\le \mathbb{E}_{\Sigma,\mathcal{D}}\Big[ \big| f_{\mathbf{W}_0 + \mathbf{W}\Sigma}(\mathbf{x}) - f_{\mathbf{W}}^Q(\mathbf{x}) \big| \Big] \\
&\le \left( \mathbb{E}_{\Sigma,\mathcal{D}}\Big[ (f_{\mathbf{W}\Sigma}^L(\mathbf{x}) + \Delta_{\mathbf{W}\Sigma}^Q(\mathbf{x}))^2 \Big] \right)^{1/2} \\
&\le \left( 2\mathbb{E}_{\Sigma,\mathcal{D}}\big[ (f_{\mathbf{W}\Sigma}^L(\mathbf{x}))^2 \big] + 2\mathbb{E}_{\Sigma,\mathcal{D}}\Big[ (\Delta_{\mathbf{W}\Sigma}^Q(\mathbf{x}))^2 \Big] \right)^{1/2} \\
&= \widetilde{O}\Big( B_x^3 \|\mathbf{W}\|_{2,4}^3 \, m^{-1/4} + d^2 B_x^2 \|\mathbf{W}\|_{2,4}^2 \, m^{-1/2} \Big),
\end{aligned}
$$

where the last step uses Lemma 11.

$\square$

**Lemma 13** (Closeness of directional gradients). *We have*

$$
\begin{aligned}
&\big| \langle \nabla L(\mathbf{W}), \mathbf{W} \rangle - \langle \nabla L^Q(\mathbf{W}), \mathbf{W} \rangle \big| \\
&\le \widetilde{O}\Big( \big( dB_x \|\mathbf{W}\|_{2,4} + \sqrt{d}B_x^5 \|\mathbf{W}\|_{2,4}^5 + B_x^3 \|\mathbf{W}\|_{2,4}^3 \big) m^{-1/4} + d^{2.5}B_x^4 \|\mathbf{W}\|_{2,4}^4 \, m^{-1/2} \Big).
\end{aligned}
$$

*Proof.* Differentiating $L$ and $L^Q$ and taking the inner product with $\mathbf{W}$, we get

$$
\langle \nabla L(\mathbf{W}), \mathbf{W} \rangle = \mathbb{E}_{\Sigma,\mathcal{D}} \left[ \ell'(y, f_{\mathbf{W}_0 + \mathbf{W}\Sigma}(\mathbf{x})) \cdot \frac{1}{\sqrt{m}} \sum_{r \le m} a_r \sigma'((\mathbf{w}_{0,r} + \Sigma_{rr}\mathbf{w}_r)^\top \mathbf{x})(\Sigma_{rr}\mathbf{w}_r^\top \mathbf{x}) \right],
$$

and

$$
\langle \nabla L^Q(\mathbf{W}), \mathbf{W} \rangle = \mathbb{E}_{\mathcal{D}} \left[ \ell'(y, f_{\mathbf{W}}^Q(\mathbf{x})) \cdot \frac{1}{\sqrt{m}} \sum_{r \le m} a_r \sigma''(\mathbf{w}_{0,r}^\top \mathbf{x}) \cdot (\mathbf{w}_r^\top \mathbf{x})^2 \right].
$$

Therefore, by expanding $\sigma'((\mathbf{w}_{0,r} + \Sigma_{rr}\mathbf{w}_r)^\top \mathbf{x})$ and noticing that $\Sigma_{rr}^2 \equiv 1$, we have

$$
|\langle \nabla L(\mathbf{W}) - \nabla L^Q(\mathbf{W}), \mathbf{W} \rangle| = \left| \underbrace{\mathbb{E}_{\boldsymbol{\Sigma},\mathcal{D}} \left[ \ell'(y, f_{\mathbf{W}_0 + \mathbf{W}\boldsymbol{\Sigma}}(\mathbf{x})) \cdot \frac{1}{\sqrt{m}} \sum_{r \leq m} a_r \sigma'(\mathbf{w}_{0,r}^\top \mathbf{x})(\Sigma_{rr}\mathbf{w}_r^\top \mathbf{x}) \right]}_{\text{I}} \right.
$$

$$
+ \underbrace{\mathbb{E}_{\boldsymbol{\Sigma},\mathcal{D}} \left[ \left( \ell'(y, f_{\mathbf{W}_0 + \mathbf{W}\boldsymbol{\Sigma}}(\mathbf{x})) - \ell'(y, f_{\mathbf{W}}^Q(\mathbf{x})) \right) \cdot \frac{1}{\sqrt{m}} \sum_{r \leq m} a_r \sigma''(\mathbf{w}_{0,r}^\top \mathbf{x}) \cdot (\mathbf{w}_r^\top \mathbf{x})^2 \right]}_{\text{II}}
$$

$$
+ \underbrace{\mathbb{E}_{\boldsymbol{\Sigma},\mathcal{D}} \left[ \ell'(y, f_{\mathbf{W}_0 + \mathbf{W}\boldsymbol{\Sigma}}(\mathbf{x})) \right.}_{}
$$

$$
\left. \left. \underbrace{\cdot \frac{1}{\sqrt{m}} \sum_{r \leq m} a_r \left( \sigma'((\mathbf{w}_{0,r} + \Sigma_{rr}\mathbf{w}_r)^\top \mathbf{x}) - \sigma'(\mathbf{w}_{0,r}^\top \mathbf{x}) - \sigma''(\mathbf{w}_{0,r}^\top \mathbf{x})(\Sigma_{rr}\mathbf{w}_r^\top \mathbf{x}) \right)(\Sigma_{rr}\mathbf{w}_r^\top \mathbf{x}) \right]}_{\text{III}} \right|.
$$

We now bound the three terms separately. Recall that $|\ell'| \leq 1$ and $\ell'(y, z)$ is 1-Lipschitz in $z$. For term I we have by Cauchy-Schwarz that

$$
|\text{I}| \leq \left( \mathbb{E}_{\mathcal{D}} \left[ \frac{1}{m} \sum_{r \leq m} a_r^2 \sigma'(\mathbf{w}_{0,r}^\top \mathbf{x})^2 (\mathbf{w}_r^\top \mathbf{x})^2 \right] \right)^{1/2}
$$

$$
\leq \left( C \max_{r \in [m], i \in [n]} (\mathbf{w}_{0,r}^\top \mathbf{x}_i)^4 \cdot \frac{1}{m} \sum_{r \leq m} \|\mathbf{w}_r\|_2^2 B_x^2 \right)^{1/2}
$$

$$
\leq \widetilde{O}(dB_x) \cdot \left( \frac{1}{m} \sum_{r \leq m} \|\mathbf{w}_r\|_2^4 \right)^{1/4} = O\left( dB_x \|\mathbf{W}\|_{2,4} m^{-1/4} \right).
$$

For term II, we have

$$
|\text{II}| \overset{(i)}{\leq} \left( \mathbb{E}_{\boldsymbol{\Sigma},\mathcal{D}} \left[ (f_{\mathbf{W}_0 + \mathbf{W}\boldsymbol{\Sigma}}(\mathbf{x}) - f_{\mathbf{W}}^Q(\mathbf{x}))^2 \right] \right)^{1/2} \cdot \left( \mathbb{E}_{\mathcal{D}} \left[ (f_{\mathbf{W}}^Q(\mathbf{x}))^2 \right] \right)^{1/2}
$$

$$
\overset{(ii)}{\leq} \widetilde{O}\left( B_x^3 \|\mathbf{W}\|_{2,4}^3 m^{-1/4} + d^2 B_x^2 \|\mathbf{W}\|_{2,4}^2 m^{-1/2} \right) \cdot \widetilde{O}(\sqrt{d} B_x^2 \|\mathbf{W}\|_{2,4}^2)
$$

$$
= \widetilde{O}\left( \sqrt{d} B_x^5 \|\mathbf{W}\|_{2,4}^5 m^{-1/4} + d^{2.5} B_x^4 \|\mathbf{W}\|_{2,4}^4 m^{-1/2} \right).
$$

where (i) uses Cauchy-Schwarz and (ii) uses the bounds in Lemma 10 and 11. For term III we first note by the smoothness of $\sigma'$ that

$$
\left| a_r \left( \sigma'((\mathbf{w}_{0,r} + \Sigma_{rr}\mathbf{w}_r)^\top \mathbf{x}) - \sigma'(\mathbf{w}_{0,r}^\top \mathbf{x}) - \sigma''(\mathbf{w}_{0,r}^\top \mathbf{x})(\Sigma_{rr}\mathbf{w}_r^\top \mathbf{x}) \right)(\Sigma_{rr}\mathbf{w}_r^\top \mathbf{x}) \right|
$$

$$
\leq C \left| \Sigma_{rr}\mathbf{w}_r^\top \mathbf{x} \right|^3 \leq C B_x^3 \|\mathbf{w}_r\|_2^3.
$$

Substituting this bound into term III yields

$$
|\text{III}| \leq \frac{1}{\sqrt{m}} \sum_{r \leq m} C B_x^3 \|\mathbf{w}_r\|_2^3 \leq C\sqrt{m} B_x^3 \cdot \frac{1}{m} \sum_{r \leq m} \|\mathbf{w}_r\|_2^3
$$

$$
\leq C\sqrt{m} B_x^3 \cdot \left( \frac{1}{m} \sum_{r \leq m} \|\mathbf{w}_r\|_2^4 \right)^{3/4} = O\left( B_x^3 \|\mathbf{W}\|_{2,4}^3 m^{-1/4} \right).
$$

Putting together the bounds for term I, II, III gives the desired result. $\quad\square$

**Lemma 14** (Closeness of Hessians). *Let $\mathbf{\Sigma}'$ denote a diagonal matrix with diagonal entries drawn i.i.d. from* $\mathrm{Unif}\{\pm 1\}$. *We have for all* $\mathbf{W}, \mathbf{W}_\star \in \mathbb{R}^{d \times m}$ *that*

$$\left| \mathbb{E}_{\mathbf{\Sigma}'}\left[ (\nabla^2 L(\mathbf{W}) - \nabla^2 L^Q(\mathbf{W}))[\mathbf{W}_\star \mathbf{\Sigma}', \mathbf{W}_\star \mathbf{\Sigma}'] \right] \right|$$

$$\leq \widetilde{O}\Bigg( \left( B_x^3 \|\mathbf{W}\|_{2,4} + \sqrt{d} B_x^5 \|\mathbf{W}\|_{2,4}^3 \right) \|\mathbf{W}_\star\|_{2,4}^2 \, m^{-1/4}$$

$$+ \left( d^{2.5} B_x^4 \|\mathbf{W}\|_{2,4}^2 \|\mathbf{W}_\star\|_{2,4}^2 + B_x^2 (d^2 + \|\mathbf{W}\|_{2,\infty}^4 B_x^4) \|\mathbf{W}_\star\|_{2,4}^2 \right) m^{-1/2}$$

$$+ d B_x^4 \|\mathbf{W}\|_{2,4}^2 \|\mathbf{W}_\star\|_{2,4}^2 \, m^{-1} \Bigg).$$

*Proof.* Differentiating $L$ and $L^Q$ twice on the direction $\mathbf{W}_\star \mathbf{\Sigma}'$, we get

$$\nabla^2 L(\mathbf{W})[\mathbf{W}_\star \mathbf{\Sigma}', \mathbf{W}_\star \mathbf{\Sigma}']$$

$$= \underbrace{\mathbb{E}_{\mathbf{\Sigma}, \mathbf{\Sigma}', \mathcal{D}}\left[ \ell''(y, f_{\mathbf{W}_0 + \mathbf{W}\mathbf{\Sigma}}(\mathbf{x})) \cdot \left( \frac{1}{\sqrt{m}} \sum_{r \leq m} a_r \sigma'((\mathbf{w}_{0,r} + \Sigma_{rr}\mathbf{w}_r)^\top \mathbf{x})(\Sigma_{rr}\Sigma'_{rr}\mathbf{w}_{\star,r}^\top \mathbf{x}) \right)^2 \right]}_{\mathrm{I}(L)}$$

$$+ \underbrace{\mathbb{E}_{\mathbf{\Sigma}, \mathcal{D}}\left[ \ell'(y, f_{\mathbf{W}_0 + \mathbf{W}\mathbf{\Sigma}}(\mathbf{x})) \cdot \frac{1}{\sqrt{m}} \sum_{r \leq m} a_r \sigma''((\mathbf{w}_{0,r} + \Sigma_{rr}\mathbf{w}_r)^\top \mathbf{x})(\Sigma_{rr}\Sigma'_{rr}\mathbf{w}_{\star,r}^\top \mathbf{x})^2 \right]}_{\mathrm{II}(L)},$$

and

$$\nabla^2 L^Q(\mathbf{W})[\mathbf{W}_\star \mathbf{\Sigma}', \mathbf{W}_\star \mathbf{\Sigma}'] = \underbrace{\mathbb{E}_{\mathbf{\Sigma}', \mathcal{D}}\left[ \ell''(y, f_{\mathbf{W}}^Q(\mathbf{x})) \cdot \left( \frac{1}{\sqrt{m}} \sum_{r \leq m} a_r \sigma''(\mathbf{w}_{0,r}^\top \mathbf{x})(\mathbf{w}_r^\top \mathbf{x})(\Sigma'_{rr}\mathbf{w}_{\star,r}^\top \mathbf{x}) \right)^2 \right]}_{\mathrm{I}(L^Q)}$$

$$+ \underbrace{\mathbb{E}_{\mathbf{\Sigma}', \mathcal{D}}\left[ \ell'(y, f_{\mathbf{W}}^Q(\mathbf{x})) \cdot \frac{1}{\sqrt{m}} \sum_{r \leq m} a_r \sigma''(\mathbf{w}_{0,r}^\top \mathbf{x})(\Sigma'_{rr}\mathbf{w}_{\star,r}^\top \mathbf{x})^2 \right]}_{\mathrm{II}(L^Q)}.$$

We first bound the terms $\mathrm{I}(L)$ and $\mathrm{I}(L^Q)$. We have

$$\mathrm{I}(L) = 2\mathbb{E}_{\mathbf{\Sigma}, \mathbf{\Sigma}', \mathcal{D}}\left[ \frac{1}{m} \sum_{r \leq m} a_r^2 \sigma'((\mathbf{w}_{0,r} + \Sigma_{rr}\mathbf{w}_r)^\top \mathbf{x})^2 (\mathbf{w}_{\star,r}^\top \mathbf{x})^2 \right]$$

$$\leq C \cdot \sup_{\|\mathbf{x}\|_2 = B_x} \frac{1}{m} \sum_{r \leq m} \left( (\mathbf{w}_{0,r} + \Sigma_{rr}\mathbf{w}_r)^\top \mathbf{x} \right)^4 (\mathbf{w}_{\star,r}^\top \mathbf{x})^2$$

$$\leq C B_x^2 \cdot \frac{1}{m} \sum_{r \leq m} (\widetilde{O}(d^2) + \|\mathbf{w}_r\|_2^4 B_x^4) \|\mathbf{w}_{\star,r}\|_2^2$$

$$\leq \widetilde{O}\left( B_x^2 \left( d^2 + \|\mathbf{W}\|_{2,\infty}^4 B_x^4 \right) \|\mathbf{W}_\star\|_{2,4}^2 \, m^{-1/2} \right).$$

Using similar arguments on $\mathrm{I}(L^Q)$ gives the bound

$$\mathrm{I}(L^Q) \leq \widetilde{O}\left( d B_x^4 \|\mathbf{W}\|_{2,4}^2 \|\mathbf{W}_\star\|_{2,4}^2 \, m^{-1} \right). \tag{12}$$

We now shift attention to bounding $\mathrm{II}(L) - \mathrm{II}(L^Q)$. First note that

$$\left| \Delta_r^{\sigma''}(\mathbf{x}) \right| := \left| a_r (\sigma''((\mathbf{w}_{0,r} + \Sigma_{rr}\mathbf{w}_r)^\top \mathbf{x}) - \sigma''(\mathbf{w}_{0,r}^\top \mathbf{x}))(\mathbf{w}_{\star,r}^\top \mathbf{x})^2 \right|$$

$$\leq C |\Sigma_{rr}\mathbf{w}_r^\top \mathbf{x}| \cdot (\mathbf{w}_{\star,r}^\top \mathbf{x})^2 \leq C B_x^3 \|\mathbf{w}_r\|_2 \|\mathbf{w}_{\star,r}\|_2^2.$$

Then we have, by applying the bounds in Lemma 10 and 11,

$$\left| \mathrm{II}(L) - \mathrm{II}(L^Q) \right|$$

$$= \left| \mathbb{E}_{\boldsymbol{\Sigma}, \mathcal{D}} \left[ \ell'(y, f_{\mathbf{W}_0 + \mathbf{W}\boldsymbol{\Sigma}}(\mathbf{x})) \cdot \frac{1}{\sqrt{m}} \sum_{r \leq m} \Delta_r^{\sigma''}(\mathbf{x}) \right] + \mathbb{E}_{\boldsymbol{\Sigma}, \mathcal{D}} \left[ \left( \ell'(y, f_{\mathbf{W}_0 + \mathbf{W}\boldsymbol{\Sigma}}^Q(\mathbf{x})) - \ell'(y, f_{\mathbf{W}}^Q(\mathbf{x})) \right) \cdot 2 f_{\mathbf{W}_\star}^Q(\mathbf{x}) \right] \right|$$

$$\leq C \cdot \frac{1}{\sqrt{m}} \sum_{r \leq m} B_x^3 \left\| \mathbf{w}_r \right\|_2 \left\| \mathbf{w}_{\star, r} \right\|_2^2 + C \sqrt{\mathbb{E}\left[ \left( f_{\mathbf{W}\boldsymbol{\Sigma}}^L(\mathbf{x}) + \Delta_{\mathbf{W}\boldsymbol{\Sigma}}^Q(\mathbf{x}) \right)^2 \right]} \cdot \left( \mathbb{E}\left[ f_{\mathbf{W}_\star}^Q(\mathbf{x})^2 \right] \right)^{1/2}$$

$$\leq \widetilde{O}\left( B_x^3 \left\| \mathbf{W} \right\|_{2,4} \left\| \mathbf{W}_\star \right\|_{2,4}^2 m^{-1/4} \right) + \widetilde{O}\left( B_x^3 \left\| \mathbf{W} \right\|_{2,4}^3 m^{-1/4} + d^2 B_x^2 \left\| \mathbf{W} \right\|_{2,4}^2 m^{-1/2} \right) \cdot \widetilde{O}\left( \sqrt{d} B_x^2 \left\| \mathbf{W}_\star \right\|_{2,4}^2 \right).$$

$$= \widetilde{O}\left( \left( B_x^3 \left\| \mathbf{W} \right\|_{2,4} + \sqrt{d} B_x^5 \left\| \mathbf{W} \right\|_{2,4}^3 \right) \left\| \mathbf{W}_\star \right\|_{2,4}^2 m^{-1/4} + d^{2.5} B_x^4 \left\| \mathbf{W} \right\|_{2,4}^2 \left\| \mathbf{W}_\star \right\|_{2,4}^2 m^{-1/2} \right).$$

Combining all the bounds gives the desired result. □

### B.4 PROOF OF THEOREM 2

We apply Lemma 12, 13, and 14 to connect the neural net loss $L$ to the "clean risk" $L^Q$. First, by Lemma 12, we have for all the assumed $\mathbf{W}$ that

$$\left| L(\mathbf{W}) - L^Q(\mathbf{W}) \right| \leq \widetilde{O}\left( B_x^3 \left\| \mathbf{W} \right\|_{2,4}^3 m^{-1/4} + d^2 B_x^2 \left\| \mathbf{W} \right\|_{2,4}^2 m^{-1/2} \right).$$

Therefore we have $\left| L(\mathbf{W}) - L^Q(\mathbf{W}) \right| \leq \epsilon/6$ so long as

$$m \geq \widetilde{O}\left( B_x^{12} B_w^{12} \epsilon^{-4} + d^4 B_x^4 B_w^4 \epsilon^{-2} \right). \tag{13}$$

Applying Lemma 1, we obtain that

$$\mathbb{E}_{\boldsymbol{\Sigma}'}\left[ \nabla^2 L^Q(\mathbf{W})[\mathbf{W}_\star \boldsymbol{\Sigma}', \mathbf{W}_\star \boldsymbol{\Sigma}'] \right] - \left\langle \nabla L^Q(\mathbf{W}), \mathbf{W} \right\rangle$$
$$\leq 2(L^Q(\mathbf{W}) - \mathsf{OPT}) + \epsilon/3 \leq 2(L(\mathbf{W}) - \mathsf{OPT}) + 2\epsilon/3 \tag{14}$$

provided that the error term in Lemma 1 is bounded by $\epsilon/3$, which happens when

$$m \geq \widetilde{O}\left( d B_x^4 B_w^2 B_{w,\star}^2 \epsilon^{-1} \right). \tag{15}$$

Finally, we choose $m$ sufficiently large so that

$$\left| \mathbb{E}_{\boldsymbol{\Sigma}'}\left[ (\nabla^2 L(\mathbf{W}) - \nabla^2 L^Q(\mathbf{W}))[\mathbf{W}_\star \boldsymbol{\Sigma}', \mathbf{W}_\star \boldsymbol{\Sigma}'] \right] \right| \leq \epsilon/6$$

and

$$\left| \left\langle \nabla L(\mathbf{W}) - \nabla L^Q(\mathbf{W}), \mathbf{W} \right\rangle \right| \leq \epsilon/6,$$

which combined with (14) yields the desired result. By Lemma 13 and 14, it suffices to choose $m$ such that, to satisfy the closeness of directional gradients,

$$m \geq \widetilde{O}\left( \left( d^4 B_x^4 B_w^4 + d^2 B_x^{20} B_w^{20} + B_x^{12} B_w^{12} \right) \epsilon^{-4} + d^5 B_x^8 B_w^8 \epsilon^{-2} \right), \tag{16}$$

and to satisfy the closeness of Hessian quadratic forms,

$$m \geq O\Bigg( \left[ B_x^{12} B_w^4 B_{w,\star}^8 + d^2 B_x^{20} B_w^{12} B_{w,\star}^8 \right] \epsilon^{-4}$$
$$+ \left[ d^5 B_x^8 B_w^4 B_{w,\star}^4 + d^4 B_x^4 B_{w,\star}^4 + B_x^{12} B_w^8 B_{w,\star}^4 \right] \epsilon^{-2} + d B_x^4 B_w^2 B_{w,\star}^2 \epsilon^{-1} \Bigg). \tag{17}$$

Collecting the requirements on $m$ in (13), (15), (16), (17) and merging terms using $\epsilon \leq 1$ and $B_{w,\star} \leq B_w$, the desired result holds whenever

$$m \geq O\left( \left[ B_x^{12} B_w^{12} + d^4 B_x^4 B_w^4 + d^2 B_x^{20} B_w^{20} \right] \epsilon^{-4} + d^5 B_x^8 B_w^8 \epsilon^{-2} \right).$$

This completes the proof. □

## B.5    PROOF OF COROLLARY 3

*Proof.* For all $\lambda \geq 0$ define
$$A_\lambda := \mathbb{E}_{\boldsymbol{\Sigma}'}\left[\nabla^2 L_\lambda(\mathbf{W})[\mathbf{W}_\star \boldsymbol{\Sigma}', \mathbf{W}_\star \boldsymbol{\Sigma}']\right] - \langle \nabla L_\lambda(\mathbf{W}), \mathbf{W}\rangle + 2(L_\lambda(\mathbf{W}) - \mathsf{OPT}),$$
By Lemma 2, it suffices to show that
$$A_\lambda - A_0 \leq C\lambda \|\mathbf{W}_\star\|_{2,4}^8 - \lambda \|\mathbf{W}\|_{2,4}^8$$
for some absolute constant $C$.

Recall that $L_\lambda(\mathbf{W}) = L(\mathbf{W}) + \lambda \|\mathbf{W}_\star\|_{2,4}^8$. By differentiating $\mathbf{A} \mapsto \|\mathbf{A}\|_{2,4}^8$ we get
$$\langle \nabla(L_\lambda - L)(\mathbf{W}), \mathbf{W}\rangle = 2 \|\mathbf{W}\|_{2,4}^4 \cdot \sum_{r \leq m} \left\langle 4\lambda \|\mathbf{w}_r\|_2^2 \mathbf{w}_r, \mathbf{w}_r \right\rangle = 8\lambda \|\mathbf{W}\|_{2,4}^8$$
and
$$\nabla^2(L_\lambda - L)(\mathbf{W})[\mathbf{W}_\star \boldsymbol{\Sigma}', \mathbf{W}_\star \boldsymbol{\Sigma}']$$
$$= 8\lambda \|\mathbf{W}\|_{2,4}^4 \left[\sum_{r \leq m} \|\mathbf{w}_r\|_2^2 \|\mathbf{w}_{\star,r}\Sigma'_{rr}\|_2^2 + 2\langle \mathbf{w}_r, \mathbf{w}_{\star,r}\Sigma'_{rr}\rangle^2\right] + 32\lambda \|\mathbf{W}\|_{2,4}^4 \cdot \left(\sum_{r \leq m} \|\mathbf{w}_r\|_2^2 \langle \mathbf{w}_r, \mathbf{w}_{\star,r}\Sigma'_{rr}\rangle\right)^2$$
$$\leq 56\lambda \|\mathbf{W}\|_{2,4}^4 \sum_{r \leq m} \|\mathbf{w}_r\|_2^2 \|\mathbf{w}_{\star,r}\|_2^2 \overset{(i)}{\leq} 56\lambda \|\mathbf{W}\|_{2,4}^6 \|\mathbf{W}_\star\|_{2,4}^2 \overset{(ii)}{\leq} 14\lambda\alpha \|\mathbf{W}\|_{2,4}^8 + \frac{378\lambda}{\alpha^3} \|\mathbf{W}_\star\|_{2,4}^8,$$
where (i) used Cauchy-Schwarz and (ii) used the AM-GM inequality $p^3 q \leq \alpha p^4/4 + 27q^4/(4\alpha^3)$ for all $p, q$ and $\alpha > 0$. Substituting the above expressions into $A_\lambda - A_0$ yields
$$A_\lambda - A_0$$
$$\leq 14\lambda\alpha \|\mathbf{W}\|_{2,4}^8 + \frac{378\lambda}{\alpha^3} \|\mathbf{W}_\star\|_{2,4}^8 - 8\lambda \|\mathbf{W}\|_{2,4}^8 + 2\lambda \|\mathbf{W}\|_{2,4}^8$$
$$= (14\lambda\alpha - 6\lambda) \|\mathbf{W}\|_{2,4}^8 + \frac{378\lambda}{\alpha^3} \|\mathbf{W}_\star\|_{2,4}^8.$$
Choosing $\alpha = 5/14$ gives the desired result. $\square$

## B.6    PROOF OF THEOREM 4

We begin by choosing the regularization strength as
$$\lambda = \lambda_0 B_{w,\star}^{-8},$$
where $\lambda_0$ is a constant to be determined. Let $\epsilon$ be an accuracy parameter also to be determined.

**Localizing second-order stationary points**    We first argue that any second order stationary point $\mathbf{W}$ has to satisfy $\|\mathbf{W}\|_{2,4} \leq B_{w,0}$ for some large but controlled $B_{w,0}$. We first note that for the clean risk $L^Q$, we have for any $\mathbf{W} \in \mathbb{R}^{d \times m}$ that
$$\langle \nabla L^Q(\mathbf{W}), \mathbf{W}\rangle = \mathbb{E}_\mathcal{D}\left[\ell'(y, f_\mathbf{W}^Q(\mathbf{x})) \cdot 2f_\mathbf{W}^Q(\mathbf{x})\right]$$
$$= 2\mathbb{E}_\mathcal{D}\left[\ell'(y, f_\mathbf{W}^Q(\mathbf{x})) \cdot (f_\mathbf{W}^Q(\mathbf{x}) - f_\mathbf{0}^Q(\mathbf{x}))\right] \overset{(i)}{\geq} 2(L^Q(\mathbf{W}) - L^Q(\mathbf{0})) \overset{(ii)}{\geq} -2,$$
where (i) uses convexity of $\ell$ and (ii) uses the assumption that $\ell(y, 0) \leq 1$ for all $y \in \mathcal{Y}$.

Now, applying the coupling Lemma 13, and combining with the fact that $\left\langle \nabla_\mathbf{W}(\lambda \|\mathbf{W}\|_{2,4}^8), \mathbf{W}\right\rangle = 8\lambda \|\mathbf{W}\|_{2,4}^8$, we have simultaneously for all $\mathbf{W}$ that
$$\langle \nabla L_\lambda(\mathbf{W}), \mathbf{W}\rangle$$
$$\geq \left\langle \nabla_\mathbf{W}(\lambda \|\mathbf{W}\|_{2,4}^8), \mathbf{W}\right\rangle + \langle \nabla L^Q(\mathbf{W}), \mathbf{W}\rangle - |\langle \nabla(L - L^Q)(\mathbf{W}), \mathbf{W}\rangle|$$
$$\geq 8\lambda \|\mathbf{W}\|_{2,4}^8 - 2 - \widetilde{O}\left(\left(dB_x \|\mathbf{W}\|_{2,4} + \sqrt{d}B_x^5 \|\mathbf{W}\|_{2,4}^5 + B_x^3 \|\mathbf{W}\|_{2,4}^3\right)m^{-1/4} + d^{2.5}B_x^4 \|\mathbf{W}\|_{2,4}^4 m^{-1/2}\right).$$

Therefore we see that any stationary point $\mathbf{W}$ has to satisfy

$$\|\mathbf{W}\|_{2,4} \leq B_{w,0}$$
$$:= \widetilde{O}\Big(\lambda^{-1/8} + (\lambda^{-1}dB_x m^{-1/4})^{1/7} + (\lambda^{-1}\sqrt{d}B_x^5 m^{-1/4})^{1/3} + (\lambda^{-1}B_x^3)^{1/5} + (\lambda^{-1}d^{2.5}B_x^4 m^{-1/2})^{1/4}\Big).$$

By Corollary 3, choosing $m \geq \text{poly}(\lambda_0^{-1}, d, B_{w,\star}B_x, \epsilon)$, the coupling error is bounded by $\epsilon$ in $\mathsf{B}_{2,4}(B_{w,0})$, i.e. for all $\mathbf{W} \in \mathsf{B}_{2,4}(B_{w,0})$ we have that

$$\mathbb{E}_{\boldsymbol{\Sigma}'}\left[\nabla^2 L_\lambda(\mathbf{W})[\mathbf{W}_\star\boldsymbol{\Sigma}', \mathbf{W}_\star\boldsymbol{\Sigma}']\right]$$
$$\leq \langle\nabla L_\lambda(\mathbf{W}), \mathbf{W}\rangle - 2(L_\lambda(\mathbf{W}) - \mathsf{OPT}) - \lambda\|\mathbf{W}\|_{2,4}^8 + C\lambda\|\mathbf{W}_\star\|_{2,4}^8 + \epsilon, \tag{18}$$

where $C = O(1)$ is an absolute constant.

**Bounding loss and norm**    Choosing

$$\lambda_0 = \frac{1}{C}\cdot(2\gamma\mathsf{OPT} + \epsilon),$$

we get that $C\lambda B_{w,\star}^8 = 2\gamma\mathsf{OPT} + \epsilon$, and thus the bound (18) reads

$$\mathbb{E}_{\boldsymbol{\Sigma}'}\left[\nabla^2 L_\lambda(\mathbf{W})[\mathbf{W}_\star\boldsymbol{\Sigma}', \mathbf{W}_\star\boldsymbol{\Sigma}']\right]$$
$$\leq \langle\nabla L_\lambda(\mathbf{W}), \mathbf{W}\rangle - 2(L_\lambda(\mathbf{W}) - \mathsf{OPT}) - \lambda\|\mathbf{W}\|_{2,4}^8 + 2\gamma\mathsf{OPT} + 2\epsilon.$$

For the second-order stationary point $\widehat{\mathbf{W}}$, the gradient term vanishes and the Hessian term is non-negative, so we get

$$2(L_\lambda(\widehat{\mathbf{W}}) - \mathsf{OPT}) \leq 2(\gamma\mathsf{OPT} + \epsilon) - \lambda\left\|\widehat{\mathbf{W}}\right\|_{2,4}^8 \leq 2(\gamma\mathsf{OPT} + \epsilon)$$

and thus

$$L_\lambda(\widehat{\mathbf{W}}) \leq (1+\gamma)\mathsf{OPT} + \epsilon.$$

Further, by re-writing (18), we obtain

$$\lambda\left\|\widehat{\mathbf{W}}\right\|_{2,4}^8 \leq C\lambda B_{w,\star}^8 + 2(\mathsf{OPT} - L_\lambda(\widehat{\mathbf{W}})) + \epsilon \leq C\lambda B_{w,\star}^8 + 2\mathsf{OPT} + \epsilon$$
$$\leq C\lambda B_{w,\star}^8 \cdot \left(1 + \frac{2\mathsf{OPT} + \epsilon}{2\gamma\mathsf{OPT} + \epsilon}\right) = O(1)\cdot\lambda B_{w,\star}^8,$$

for any $\gamma = O(1)$. This is the desired result. $\qquad\square$

## C  PROOFS FOR SECTION 5

### C.1  PROOF OF LEMMA 5

As the loss $\ell(y, z)$ is 1-Lipschitz in $z$ for all $y$, by the Rademacher contraction theorem (Wainwright, 2019, Chapter 5) we have that

$$
\mathbb{E}_{\boldsymbol{\sigma},\mathbf{x}}\left[\sup_{\|\mathbf{W}\|_{2,4}\leq B_w} \frac{1}{n}\sum_{i=1}^n \sigma_i\ell(y_i, f_{\mathbf{W}}^Q(\mathbf{x}_i))\right]
$$

$$
\leq 2\mathbb{E}_{\boldsymbol{\sigma},\mathbf{x}}\left[\sup_{\|\mathbf{W}\|_{2,4}\leq B_w} \frac{1}{n}\sum_{i=1}^n \sigma_i f_{\mathbf{W}}^Q(\mathbf{x}_i)\right] + \mathbb{E}_{\boldsymbol{\sigma},\mathbf{x}}\left[\frac{1}{n}\sum_{i=1}^n \sigma_i\ell(y_i,0)\right]
$$

$$
\leq \mathbb{E}_{\boldsymbol{\sigma},\mathbf{x}}\left[\sup_{\|\mathbf{W}\|_{2,4}\leq B_w} \frac{1}{\sqrt{m}}\sum_{r\leq m}\left\langle \frac{1}{n}\sum_{i=1}^n \sigma_i a_r \sigma''(\mathbf{w}_{0,r}^\top\mathbf{x}_i)\mathbf{x}_i\mathbf{x}_i^\top, \mathbf{w}_r\mathbf{w}_r^\top\right\rangle\right] + \frac{1}{\sqrt{n}}
$$

$$
\leq \mathbb{E}_{\boldsymbol{\sigma},\mathbf{x}}\left[\sup_{\|\mathbf{W}\|_{2,4}\leq B_w}\max_{r\in[m]}\left\|\frac{1}{n}\sum_{i=1}^n a_r\sigma_i\sigma''(\mathbf{w}_{0,r}^\top\mathbf{x}_i)\mathbf{x}_i\mathbf{x}_i^\top\right\|_{\text{op}}\cdot \frac{1}{\sqrt{m}}\sum_{r\leq m}\|\mathbf{w}_r\mathbf{w}_r^\top\|_*\right] + \frac{1}{\sqrt{n}}
$$

$$
\leq \mathbb{E}_{\boldsymbol{\sigma},\mathbf{x}}\left[\max_{r\in[m]}\left\|\frac{1}{n}\sum_{i=1}^n \sigma_i\sigma''(\mathbf{w}_{0,r}^\top\mathbf{x}_i)\mathbf{x}_i\mathbf{x}_i^\top\right\|_{\text{op}}\right]\cdot \underbrace{\sup_{\|\mathbf{W}\|_{2,4}\leq B_w}\frac{1}{\sqrt{m}}\sum_{r\leq m}\|\mathbf{w}_r\|_2^2}_{\leq B_w^2} + \frac{1}{\sqrt{n}},
$$

where the last step used the power mean (or Cauchy-Schwarz) inequality on $\{\|\mathbf{w}_r\|_2\}$. $\qquad\square$

### C.2  PROOF OF THEOREM 6

We first relate the generalization of $L$ to that of $L^Q$ through

$$
L_P(\widehat{\mathbf{W}}) - L(\widehat{\mathbf{W}}) \leq L_P(\widehat{\mathbf{W}}) - L_P^Q(\widehat{\mathbf{W}}) + L_P^Q(\widehat{\mathbf{W}}) - L^Q(\widehat{\mathbf{W}}) + L^Q(\widehat{\mathbf{W}}) - L(\widehat{\mathbf{W}}).
$$

By Lemma 12, we have simultaneously for all $\mathbf{W} \in \mathsf{B}_{2,4}(B_w)$ that

$$
\left|L(\mathbf{W}) - L^Q(\mathbf{W})\right| \leq \widetilde{O}\Big(B_x^3 B_w^3 m^{-1/4} + d^2 B_x^2 B_w^2 m^{-1/2}\Big). \tag{19}
$$

Further, from the proof we see that the argument does not depend on the distribution of $\mathbf{x}$ (it holds uniformly for all $\mathbf{x} \in \mathbb{S}^{d-1}(B_x)$), therefore for the population version we also have the bound

$$
\left|L_P(\mathbf{W}) - L_P^Q(\mathbf{W})\right| \leq \widetilde{O}\Big(B_x^3 B_w^3 m^{-1/4} + d^2 B_x^2 B_w^2 m^{-1/2}\Big). \tag{20}
$$

These bounds hold for all $\mathbf{W} \in \mathsf{B}_{2,4}(B_w)$ so apply to $\widehat{\mathbf{W}}$. Therefore it remains to bound $L_P^Q(\widehat{\mathbf{W}}) - L^Q(\widehat{\mathbf{W}})$, i.e. the generalization of the quadratic model.

**Generalization of quadratic model**  By symmetrization and applying Lemma 5, we have

$$
\mathbb{E}_{\mathbf{W}_0,\mathcal{D}}\Big[L_P^Q(\widehat{\mathbf{W}}) - L^Q(\widehat{\mathbf{W}})\Big] \leq \mathbb{E}_{\mathbf{W}_0,\mathcal{D}}\left[\sup_{\|\mathbf{W}\|_{2,4}\leq B_w} L_P^Q(\mathbf{W}) - L^Q(\mathbf{W})\right]
$$

$$
\leq 2\mathbb{E}_{\mathbf{W}_0,\boldsymbol{\sigma},\mathbf{x}}\left[\sup_{\|\mathbf{W}\|_{2,4}\leq B_w}\frac{1}{n}\sum_{i=1}^n \sigma_i\ell(y_i, f_{\mathbf{W}}^Q(\mathbf{x}_i))\right] \tag{21}
$$

$$
\leq 4B_w^2 \mathbb{E}_{\mathbf{W}_0,\boldsymbol{\sigma},\mathbf{x}}\left[\max_{r\in[m]}\left\|\frac{1}{n}\sum_{i=1}^n \sigma_i\sigma''(\mathbf{w}_{0,r}^\top\mathbf{x}_i)\mathbf{x}_i\mathbf{x}_i^\top\right\|_{\text{op}}\right] + \frac{2}{\sqrt{n}}.
$$

We now focus on bounding the expected max operator norm above. First, we apply the matrix concentration Lemma 8 to deduce that

$$
\mathbb{E}_{\mathbf{W}_0, \boldsymbol{\sigma}, \mathbf{x}} \left[ \max_{r \in [m]} \left\| \frac{1}{n} \sum_{i=1}^{n} \sigma_i \sigma''(\mathbf{w}_{0,r}^\top \mathbf{x}_i) \mathbf{x}_i \mathbf{x}_i^\top \right\|_{\mathrm{op}} \right]
$$

$$
\leq 4\sqrt{\log(2dm)} \cdot \mathbb{E}_{\mathbf{W}_0, \mathbf{x}} \left[ \sqrt{ \max_{r \in [m]} \left\| \frac{1}{n^2} \sum_{i=1}^{n} \sigma''(\mathbf{w}_{0,r}^\top \mathbf{x}_i)^2 \left\| \mathbf{x}_i \right\|_2^2 \mathbf{x}_i \mathbf{x}_i^\top \right\|_{\mathrm{op}} } \right]
$$

$$
\leq 4 B_x \sqrt{ \frac{\log(2dm)}{n} } \cdot \mathbb{E}_{\mathbf{W}_0, \mathbf{x}} \left[ \sqrt{ \max_{r,i} \sigma''(\mathbf{w}_{0,r}^\top \mathbf{x}_i)^2 \cdot \left\| \frac{1}{n} \sum_{i=1}^{n} \mathbf{x}_i \mathbf{x}_i^\top \right\|_{\mathrm{op}} } \right]
$$

$$
\leq 4 B_x \sqrt{ \frac{\log(2dm)}{n} } \left( \mathbb{E}_{\mathbf{W}_0, \mathbf{x}} \left[ \max_{r,i} \sigma''(\mathbf{w}_{0,r}^\top \mathbf{x}_i)^2 \right] \cdot \mathbb{E}_{\mathbf{x}} \left[ \left\| \frac{1}{n} \sum_{i=1}^{n} \mathbf{x}_i \mathbf{x}_i^\top \right\|_{\mathrm{op}} \right] \right)^{1/2}
$$

As $|\sigma''(t)| \leq Ct$ and $\mathbf{w}_{0,r}^\top \mathbf{x}_i \sim \mathsf{N}(0,1)$ for all $(r,i)$, by standard expected max bound on sub-exponential variables we have

$$
\mathbb{E}_{\mathbf{W}_0, \mathbf{x}} \left[ \max \sigma''(\mathbf{w}_{0,r}^\top \mathbf{x}_i)^2 \right] \leq O(\log(mn)) = \widetilde{O}(1).
$$

Therefore defining

$$
M_{x,\mathrm{op}} := \left( B_x^{-2} \cdot \mathbb{E}_{\mathbf{x}} \left[ \frac{1}{n} \sum_{i=1}^{n} \mathbf{x}_i \mathbf{x}_i^\top \right] \right)^{1/2},
$$

and substituting the above bound into (21) yields that

$$
\mathbb{E}_{\mathbf{W}_0, \mathcal{D}} \left[ L_P^Q(\widehat{\mathbf{W}}) - L^Q(\widehat{\mathbf{W}}) \right] \leq \widetilde{O} \left( \frac{B_x^2 B_w^2 M_{x,\mathrm{op}}}{\sqrt{n}} + \frac{1}{\sqrt{n}} \right).
$$

Combining the bound with the coupling error (19) and (20), we arrive at the desired result.

For $M_{x,\mathrm{op}}$ we have two versions of bounds:

(a) We always have $\left\| \sum_{i \leq n} \mathbf{x}_i \mathbf{x}_i^\top / n \right\|_{\mathrm{op}} \leq B_x^2$, and thus $M_{x,\mathrm{op}} \leq 1$.

(b) If, in addition, $\mathbf{x}$ is uniformly distributed on the sphere $\mathbb{S}^{d-1}(B_x)$ or the hypercube $\left\{ \pm B_x / \sqrt{d} \right\}^d$, then we have by standard covariance concentration (Vershynin, 2018, Theorem 4.7.1) that $\mathbb{E}_{\mathbf{x}} \left[ \left\| \sum_{i \leq n} \mathbf{x}_i \mathbf{x}_i^\top / n \right\|_{\mathrm{op}} \right] \leq B_x^2 / d \cdot O(1 + \sqrt{d/n} + d/n) = O(B_x^2/d)$ when $n \geq d$. More generally, if for all $\mathbf{v} \in \mathbb{S}^{d-1}(1)$ we have

$$
\left\| \mathbf{v}^\top \mathbf{x} \right\|_{\psi_2} \leq K \sqrt{\mathbf{v}^\top \mathrm{Cov}(\mathbf{x}) \mathbf{v}},
$$

and that $\kappa(\mathrm{Cov}(\mathbf{x})) \leq \kappa$, then we have $\left\| \mathrm{Cov}(\mathbf{x})) \right\|_{\mathrm{op}} \leq \kappa B_x^2 / d$. Applying (Vershynin, 2018, Theorem 4.7.1), we get $M_{x,\mathrm{op}} \leq \kappa / \sqrt{d}$ whenever $n \geq O(K^4 d)$.

$\square$

## C.3 EXPRESSIVE POWER OF INFINITELY WIDE QUADRATIC MODELS

**Lemma 15** (Expressivity of $f^Q$ with infinitely many neurons). *Suppose $f_\star(\mathbf{x}) = \alpha(\boldsymbol{\beta}^\top \mathbf{x})^p$ for some $\alpha \in \mathbb{R}$, $\boldsymbol{\beta} \in \mathbb{R}^d$, and $p \geq 2$ and such that $p - 2 \in \{1\} \cup \{2\ell\}_{\ell \geq 0}$. Suppose further that we use $\sigma(t) = \frac{1}{6} \mathrm{relu}^3(t)$ (so that $\sigma''(t) = \mathrm{relu}(t)$), then there exists choices of $(\mathbf{w}_+, \mathbf{w}_-)$ that depends on $\mathbf{w}_0$ such that*

$$
\mathbb{E}_{\mathbf{w}_0} \left[ \sigma''(\mathbf{w}_0^\top \mathbf{x}) \left( (\mathbf{w}_+^\top \mathbf{x})^2 - (\mathbf{w}_-^\top \mathbf{x})^2 \right) \right] = f_\star(\mathbf{x})
$$

*and further satisfies the norm bound*

$$
\mathbb{E}_{\mathbf{w}_0} \left[ \left\| \mathbf{w}_+ \right\|_2^4 + \left\| \mathbf{w}_- \right\|_2^4 \right] \leq 2\pi((p-2) \vee 1)^3 \alpha^2 B_x^{2(p-2)} \left\| \boldsymbol{\beta} \right\|_2^{2p}.
$$

*Proof.* Our proof builds on reducing the problem from representing $(\boldsymbol{\beta}^\top \mathbf{x})^p$ via quadratic networks to representing $(\boldsymbol{\beta}^\top \mathbf{x})^{p-2}$ through a random feature model. More precisely, we consider choosing

$$(\mathbf{w}_+, \mathbf{w}_-) = \left( \sqrt{[a]_+} \cdot \boldsymbol{\beta}, \sqrt{[a]_-} \cdot \boldsymbol{\beta} \right), \tag{22}$$

where $a$ is a real-valued random scalar that can depend on $\mathbf{w}_0$, and $\boldsymbol{\beta}$ is the fixed coefficient vector in $f_\star$. With this choice, the quadratic network reduces to

$$\mathbb{E}_{\mathbf{w}_0} \left[ \sigma''(\mathbf{w}_0^\top \mathbf{x}) \big( (\mathbf{w}_+^\top \mathbf{x})^2 - (\mathbf{w}_-^\top \mathbf{x})^2 \big) \right]$$
$$= \mathbb{E}_{\mathbf{w}_0} \left[ \sigma''(\mathbf{w}_0^\top \mathbf{x}) \big( a_+ (\boldsymbol{\beta}^\top \mathbf{x})^2 - a_- (\boldsymbol{\beta}^\top \mathbf{x})^2 \big) \right] = (\boldsymbol{\beta}^\top \mathbf{x})^2 \mathbb{E}_{\mathbf{w}_0} \left[ \sigma''(\mathbf{w}_0^\top \mathbf{x}) a \right].$$

Therefore, to let the above express $f_\star(\mathbf{x}) = \alpha(\boldsymbol{\beta}^\top \mathbf{x})^p$, it suffices to choose $a$ such that

$$\mathbb{E}[\sigma''(\mathbf{w}_0^\top \mathbf{x}) a] \equiv \alpha(\boldsymbol{\beta}^\top \mathbf{x})^{p-2} \tag{23}$$

for all $\mathbf{x}$. By Lemma 9, there exists $a = a(\mathbf{w}_0)$ satisfying (23) and such that

$$\mathbb{E}_{\mathbf{w}_0}[a^2] \leq 2\pi((p-2) \vee 1)^3 \alpha^2 B_x^{2(p-2)} \|\boldsymbol{\beta}\|_2^{2(p-2)} .$$

Using this $a$ in (22), the quadratic network induced by $(\mathbf{w}_+, \mathbf{w}_-)$ has the desired expressivity, and further satisfies the expected 4th power norm bound

$$\mathbb{E}_{\mathbf{w}_0}[\|\mathbf{w}_+\|_2^4 + \|\mathbf{w}_-\|_2^4]$$
$$= \mathbb{E}_{\mathbf{w}_0}[[a]_+^2 + [a]_-^2] \cdot \|\boldsymbol{\beta}\|_2^4 = \mathbb{E}_{\mathbf{w}_0}[a^2] \|\boldsymbol{\beta}\|_2^4 \leq 2\pi((p-2) \vee 1)^3 \alpha^2 B_x^{2(p-2)} \|\boldsymbol{\beta}\|_2^{2p} .$$

This is the desired result.

$\square$

## C.4 Proof of Theorem 7

We begin by stating and proving the result for $k = 1$ in Appendix C.4.1, i.e. when $f_\star = \alpha(\boldsymbol{\beta}^\top \mathbf{x})^p$ is a single "one-directional" polynomial. The main theorem then follows as a straightforward extension of the $k = 1$ case, which we prove in Appendix C.4.2.

### C.4.1 Expressing a single "one-directional" polynomial

**Theorem 16** (Expressivity of $f^Q$). *Suppose $\{(a_r, \mathbf{w}_{0,r})\}$ are generated according to the symmetric initialization (3), and $f_\star(\mathbf{x}) = \alpha(\boldsymbol{\beta}^\top \mathbf{x})^p$ where $p - 2 \in \{1\} \cup \{2\ell\}_{\ell \geq 0}$. Suppose further that we use $\sigma(t) = \frac{1}{6}\mathrm{relu}^3(t)$ (so that $\sigma''(t) = \mathrm{relu}(t)$), then so long as the width is sufficiently large:*

$$m \geq \widetilde{O}\big( ndp^3 \alpha^2 (B_x \|\boldsymbol{\beta}\|_2)^{2p} \epsilon^{-2} \big),$$

*we have with probability at least $1 - \delta$ (over $\mathbf{W}_0$) that there exists $\mathbf{W}_\star \in \mathbb{R}^{d \times m}$ such that*

$$\big| L^Q(\mathbf{W}_\star) - L(f_\star) \big| \leq \epsilon \quad \text{and} \quad \|\mathbf{W}_\star\|_{2,4}^4 \leq B_{w,\star}^4 = O\Big( p^3 \alpha^2 B_x^{2(p-2)} \|\boldsymbol{\beta}\|_2^{2p} \delta^{-1} \Big).$$

**Proof of Theorem 16** We build on the infinite-neuron construction in Lemma 15. Given the symmetric initialization $\{\mathbf{w}_{0,r}\}_{r=1}^m$, for all $r \in [m/2]$, we consider $\mathbf{W}_\star \in \mathbb{R}^{d \times m}$ defined through

$$(\mathbf{w}_{\star,r}, \mathbf{w}_{\star,r+m/2}) = \Big( 2m^{-1/4} \mathbf{w}_+(\mathbf{w}_{0,r}), 2m^{-1/4} \mathbf{w}_-(\mathbf{w}_{0,r}) \Big),$$

where we recall $(\mathbf{w}_+(\mathbf{w}_0), \mathbf{w}_-(\mathbf{w}_0)) = (\sqrt{a_+(\mathbf{w}_0)}\boldsymbol{\beta}, \sqrt{a_-(\mathbf{w}_0)}\boldsymbol{\beta})$. We then have

$$f_{\mathbf{W}_\star}^Q(\mathbf{x}) = \frac{1}{2\sqrt{m}} \sum_{r \leq m/2} \sigma''(\mathbf{w}_{0,r}^\top \mathbf{x}) \Big[ (\mathbf{w}_{\star,r}^\top \mathbf{x})^2 - (\mathbf{w}_{\star,r+m/2}^\top \mathbf{x})^2 \Big]$$

$$= \frac{2}{m} \sum_{r \leq m/2} \sigma''(\mathbf{w}_{0,r}^\top \mathbf{x}) \big[ (\mathbf{w}_+(\mathbf{w}_{0,r})^\top \mathbf{x})^2 - (\mathbf{w}_-(\mathbf{w}_{0,r})^\top \mathbf{x})^2 \big]$$

$$= \left[ \frac{1}{m/2} \sum_{r \leq m/2} \sigma''(\mathbf{w}_{0,r}^\top \mathbf{x}) a(\mathbf{w}_{0,r}) \right] \cdot (\boldsymbol{\beta}^\top \mathbf{x})^2.$$

**Bound on** $\|\mathbf{W}_\star\|_{2,4}$   As $f_\star(\mathbf{x}) = \alpha(\boldsymbol{\beta}^\top\mathbf{x})^p$, Lemma 15 guarantees that the coefficient $a(\mathbf{w}_0)$ involved above satisfies that

$$R_a^2 := \mathbb{E}_{\mathbf{w}_0}[a(\mathbf{w}_0)^2] \le 2\pi((p-2)\vee 1)^3\alpha^2 B_x^{2(p-2)}\|\boldsymbol{\beta}\|_2^{2(p-2)}.$$

By Markov inequality, we have with probability at least $1 - \delta/2$ that

$$\frac{1}{m/2}\sum_{r\le m}a(\mathbf{w}_{0,r})^2 \le 4\pi((p-2)\vee 1)^3\alpha^2 B_x^{2(p-2)}\|\boldsymbol{\beta}\|_2^{2(p-2)}\delta^{-1},$$

which yields the bound

$$\|\mathbf{W}\|_{2,4}^4 = \sum_{r\le m}\|\mathbf{w}_{\star,r}\|_2^4$$

$$\le \|\boldsymbol{\beta}\|_2^4\cdot\sum_{r\le m/2}16m^{-1}a(\mathbf{w}_{0,r})^2 = 8\|\boldsymbol{\beta}\|_2^4\cdot\frac{1}{m/2}\sum_{r\le m/2}a(\mathbf{w}_{0,r})^2$$

$$\le 32\pi[(p-2)^3\vee 1]\alpha^2 B_x^{2(p-2)}\|\boldsymbol{\beta}\|_2^{2p}\delta^{-1}.$$

**Concentration of function**   Let $f_m(\mathbf{x}) = \frac{1}{m}\sum_{r\le m/2}\sigma''(\mathbf{w}_{0,r}^\top\mathbf{x})a(\mathbf{w}_{0,r})$. We now show the concentration of $f_m$ to $f_{\star,p-2}(\mathbf{x}) := \alpha(\boldsymbol{\beta}^\top\mathbf{x})^{p-2}$ over the dataset $\{\mathbf{x}_1,\ldots,\mathbf{x}_n\}$. We perform a truncation argument: let $R$ be a large radius (to be chosen) satisfying

$$\mathbb{P}_{\mathbf{W}_0}\left(\sup_{r\in[m]}\|\mathbf{w}_{0,r}\|_2 \ge RB_x^{-1}\right) \ge 1 - \delta/2. \tag{24}$$

On this event we have

$$f_m(\mathbf{x}) = \frac{1}{m}\sum_{r\le m}\sigma''(\mathbf{w}_{0,r}^\top\mathbf{x})a(\mathbf{w}_{0,r})\mathbf{1}\left\{\|\mathbf{w}_{0,r}\|_2 \le RB_x^{-1}\right\} := f_m^R(\mathbf{x}).$$

Letting $f_{\star,p-2}^R(\mathbf{x}) := \mathbb{E}_{\mathbf{w}_0}[\sigma''(\mathbf{w}_0^\top\mathbf{x})a(\mathbf{w}_0)\mathbf{1}\{\|\mathbf{w}_0\|_2 \le RB_x^{-1}\}]$, we have

$$\mathbb{E}_{\mathbf{W}_0}\left[\left(f_m(\mathbf{x}) - f_{\star,p-2}^R(\mathbf{x})\right)^2\right] = \frac{1}{m}\mathbb{E}_{\mathbf{w}_0}\left[\sigma''(\mathbf{w}_0^\top\mathbf{x})a^2(\mathbf{w}_0)\mathbf{1}\{\|\mathbf{w}_0\|_2 \le R\}\right] \le C\frac{R^2R_a^2}{m}.$$

Applying Chebyshev inequality and a union bound, we get

$$\mathbb{P}\left(\max_i|f_m(\mathbf{x}_i) - f_{\star,p-2}(\mathbf{x}_i)| \ge t\right) \le C\frac{nR^2R_a^2}{mt^2}.$$

For any $\epsilon > 0$, by substituting in $t = \epsilon B_x^{-2}\|\boldsymbol{\beta}\|_2^{-2}/2$, we see that

$$m \ge O\left(nR^2R_a^2B_x^4\|\boldsymbol{\beta}\|_2^4\epsilon^{-2}\right) = O\left(nR^2(p-2)^3\alpha^2B_x^{2p}\|\boldsymbol{\beta}\|_2^{2p}\epsilon^{-2}\right) \tag{25}$$

ensures that

$$\max_{i\in[n]}|f_m(\mathbf{x}_i) - f_{\star,p-2}^R(\mathbf{x}_i)| \le \epsilon B_x^{-2}\|\boldsymbol{\beta}\|^{-2}/2. \tag{26}$$

Next, for any $\mathbf{x}$ we have the bound

$$\left|f_{\star,p-2}^R(\mathbf{x}) - f_{\star,p-2}(\mathbf{x})\right| = \left|\mathbb{E}_{\mathbf{w}_0}[\sigma''(\mathbf{w}_0^\top\mathbf{x})a(\mathbf{w}_0)\mathbf{1}\{\|\mathbf{w}_0\|_2 > R\}]\right|$$

$$\le \mathbb{E}[a(\mathbf{w}_0)^2]^{1/2}\cdot\mathbb{E}[\sigma''(\mathbf{w}_0^\top\mathbf{x})^4]^{1/4}\cdot\mathbb{P}(\|\mathbf{w}_0\|_2 > R)^{1/4}$$

$$\le R_a\cdot C/\sqrt{d}\cdot\mathbb{P}(\|\mathbf{w}_0\|_2 > R)^{1/4}.$$

Choosing $R$ such that

$$\mathbb{P}(\|\mathbf{w}_0\|_2 > R) \le c\frac{\sqrt{d}\epsilon^4}{R_aB_x^8\|\boldsymbol{\beta}\|_2^8} \tag{27}$$

ensures that

$$\max_i\left|f_{\star,p-2}^R(\mathbf{x}_i) - f_{\star,p-2}(\mathbf{x}_i)\right| \le \frac{\epsilon B_x^{-2}\|\boldsymbol{\beta}\|_2^{-2}}{2}. \tag{28}$$

Combining (35) and (37), we see that with probability at least $1 - \delta$,

$$\max_{i \in [n]} \left| f^Q_{\mathbf{W}_\star}(\mathbf{x}_i) - f_\star(\mathbf{x}_i) \right| = \max_{i \in [n]} |f_m(\mathbf{x}_i) - f_{\star, p-2}(\mathbf{x}_i)| \cdot (\boldsymbol{\beta}^\top \mathbf{x}_i)^2$$

$$\leq 2 \cdot \frac{\epsilon B_x^{-2} \|\boldsymbol{\beta}\|_2^{-2}}{2} \cdot B_x^2 \|\boldsymbol{\beta}\|_2^2 = \epsilon$$

and thus

$$\left| L^Q(\mathbf{W}_\star) - L(f_\star) \right| \leq \epsilon. \tag{29}$$

To satisfy the requirements for $m$ and $R$ in (36) and (34), we first set $R = \widetilde{O}(\sqrt{d})$ (with sufficiently large log factor) to satisfy (36) by standard Gaussian norm concentration (cf. Appendix A.3), and by (34) it suffices to set $m$ as

$$m \geq \widetilde{O}\big(nd(p - 2)^3 \alpha^2 (B_x \|\boldsymbol{\beta}\|_2)^{2p} \epsilon^{-2}\big).$$

for (38) to hold. $\qquad \square$

### C.4.2 PROOF OF MAIN THEOREM

We apply Theorem 16 $k$ times: let

$$f_{\star, j}(\mathbf{x}) := \alpha_j (\boldsymbol{\beta}_j^\top \mathbf{x})^{p_j},$$

so that $f_\star = \sum_{j \leq k} f_{\star, j}$. Associate each $j$ with an *independent* set of initialization $(\mathbf{a}_0^{(j)}, \mathbf{W}_0^{(j)})$. By Theorem 16, there exists $\mathbf{W}_\star^{(j)} \in \mathbb{R}^{d \times m_j}$, where

$$m_j = \widetilde{O}\big(ndk^2 p_j^3 \alpha_j^2 (B_x \|\boldsymbol{\beta}_j\|_2)^{2p_j} \epsilon^{-2}\big)$$

such that with probability at least $1 - \delta/k$ we have

$$\max_{i \in [n]} \left| f^Q_{\mathbf{W}_\star^{(j)}}(\mathbf{x}_i) - f_\star(\mathbf{x}_i) \right| \leq \epsilon/k$$

and the norm bound

$$\left\| \mathbf{W}_\star^{(j)} \right\|_{2,4}^4 \leq O\Big(k p_j^3 \alpha_j^2 B_x^{2(p_j - 2)} \|\boldsymbol{\beta}_j\|_2^{2p_j} \delta^{-1}\Big).$$

(Note we have slightly abused notation, so that now $\left\{ f^Q_{\mathbf{W}_\star^{(j)}} \right\}_{j \in [k]}$ use a disjoint set of initial weights $(\mathbf{a}_0^{(j)}, \mathbf{W}_0^{(j)})$.) Concatenating all the $(\mathbf{W}_\star^{(j)}, \mathbf{a}_0^{(j)}, \mathbf{W}_0^{(j)})$ and applying a union bound, we have the following: so long as the width

$$m \geq \sum_{j=1}^k m_j = \widetilde{O}\left( ndk^2 \sum_{j=1}^k p_j^3 \alpha_j^2 (B_x \|\boldsymbol{\beta}_j\|_2)^{2p_j} \epsilon^{-2} \right),$$

with probability at least $1 - \delta$ (over $\mathbf{a}_0 \in \mathbb{R}^m$ and $\mathbf{W}_0 \in \mathbb{R}^{d \times m}$), there exists $\mathbf{W}_\star \in \mathbb{R}^{d \times m}$ such that

$$\max_{i \in [n]} \left| f^Q_{\mathbf{W}_\star}(\mathbf{x}_i) - f_\star(\mathbf{x}_i) \right| \leq \epsilon,$$

which by the 1-Lipschitzness of the loss implies that

$$L^Q(\mathbf{W}_\star) \leq L(f_\star) + \epsilon = \epsilon_0 + \epsilon.$$

Further, as $\mathbf{W}_\star$ is the concatenation of $\left\{ \mathbf{W}_\star^{(j)} \right\}_{j \in [k]}$, we have the norm bound

$$\|\mathbf{W}_\star\|_{2,4}^4 = \sum_{j=1}^k \left\| \mathbf{W}_\star^{(j)} \right\|_{2,4}^4 = \widetilde{O}\left( k \sum_{j=1}^k p_j^3 \alpha_j^2 B_x^{2(p_j - 2)} \|\boldsymbol{\beta}_j\|_2^{2p_j} \delta^{-1} \right).$$

This is the desired result. $\qquad \square$

# D  EXISTENCE, GENERALIZATION, AND EXPRESSIVITY OF HIGHER-ORDER NTKS

In this section we formally study the generalization and expressivity of higher-order NTKs outlined in Section 6. Let $k \geq 2$ be an integer, and recall for any $\mathbf{W} \in \mathbb{R}^{d \times 2m}$ the definition of the $k$-th order NTK

$$f_{\mathbf{W}_0, \mathbf{W}}^{(k)}(\mathbf{x}) = \frac{1}{\sqrt{m}} \sum_{r \leq m} \frac{\sigma^{(k)}(\mathbf{w}_{0,r}^\top \mathbf{x})}{k!} \left[ (\mathbf{w}_{+,r}^\top \mathbf{x})^k - (\mathbf{w}_{-,r}^\top \mathbf{x})^k \right], \tag{30}$$

## D.1  COUPLING $f$ AND $f^{(k)}$ VIA RANDOMZIATION

Recall that for analytic $\sigma$ we have the expansion

$$f_{\mathbf{W}_0 + \mathbf{W}}(\mathbf{x}) = \frac{1}{\sqrt{m}} \sum_{r \leq m} \sigma((\mathbf{w}_{0,r} + \mathbf{w}_{+,r})^\top \mathbf{x}) - \sigma((\mathbf{w}_{0,r} + \mathbf{w}_{-,r})^\top \mathbf{x}) = \sum_{k=0}^{\infty} f_{\mathbf{W}_0, \mathbf{W}}^{(k)}(\mathbf{x}),$$

For an arbitrary $\mathbf{W}$ such that $\|\mathbf{w}_{+,r}\|_2, \|\mathbf{w}_{-,r}\|_2 = o_m(1)$, we expect that $f^{(1)}(\mathbf{x})$ is the dominating term in the expansion.

**Extracting the $k$-th order term**  We now describe an approach to finding $\mathbf{W}$ so that

$$f_{\mathbf{W}_0 + \mathbf{W}}(\mathbf{x}) = f_{\mathbf{W}_0, \mathbf{W}}^{(k)}(\mathbf{x}) + o_m(1),$$

that is, the neural net is approximately the $k$-th order NTK plus an error term that goes to zero as $m \to \infty$, thereby "escaping" the NTK regime. Our approach builds on the following randomization technique: let $z_+, z_-$ be two random variables (distributions) such that

$$\mathbb{E}[z_+^j] = \mathbb{E}[z_-^j] \text{ for } j = 0, 1, \ldots, k-1 \text{ and } \mathbb{E}[z_+^k] = \mathbb{E}[z_-^k] + 1.$$

Set $(\mathbf{w}_{+,r}, \mathbf{w}_{-,r}) = (z_{+,r} \mathbf{w}_{\star,r}, z_{-,r} \mathbf{w}_{\star,r})$, and take $\|\mathbf{w}_{\star,r}\|_2 = O(m^{-1/2k})$, we have

$$f_{\mathbf{W}_0, \mathbf{W}}^{(j)}(\mathbf{x}) = \frac{1}{\sqrt{m}} \sum_{r \leq m} \frac{1}{j!} \sigma^{(j)}(\mathbf{w}_{0,r}^\top \mathbf{x}) \underbrace{(z_{+,r}^j - z_{-,r}^j)}_{\text{mean zero}} \underbrace{(\mathbf{w}_{\star,r}^\top \mathbf{x})^j}_{O(m^{-j/2k})} = O_p(m^{-j/2k})$$

for all $j = 1, \ldots, k-1$, and

$$f_{\mathbf{W}_0, \mathbf{W}}^{(k)}(\mathbf{x}) = \frac{1}{\sqrt{m}} \sum_{r \leq m} \frac{1}{k!} \sigma^{(k)}(\mathbf{w}_{0,r}^\top \mathbf{x}) \underbrace{(z_{+,r}^k - z_{-,r}^k)}_{\text{mean=1}} \underbrace{(\mathbf{w}_{\star,r}^\top \mathbf{x})^k}_{O(m^{-1/2})} = O_P(1),$$

and

$$f_{\mathbf{W}_0, \mathbf{W}}^{(k+1)}(\mathbf{x}) = \frac{1}{\sqrt{m}} \sum_{r \leq m} \frac{1}{(k+1)!} \sigma^{(k+1)}(\mathbf{w}_{0,r}^\top \mathbf{x}) (z_{+,r}^{k+1} - z_{-,r}^{k+1}) \underbrace{(\mathbf{w}_{\star,r}^\top \mathbf{x})^{k+1}}_{O(m^{-(k+1)/2k})} = O_P(m^{-1/2k}).$$

Therefore, with high probability, all $f^{(1)}, \ldots, f^{(k-1)}$ as well as the remainder term $f - \sum_{j \leq k} f^{(j)}$ has order $O(m^{-1/2k})$, and the $k$-th order NTK $f^{(k)}$ can express an $O(1)$ function.

## D.2  GENERALIZATION AND EXPRESSIVITY OF $f^{(k)}$

We now turn to studying the generalization and expressivity of the $k$-th order NTK $f^{(k)}$, Throughout this subsection, we assume (for convenience) that

$$\sigma_k(t) := \frac{1}{k!} \sigma^{(k)}(t) \equiv \text{relu}(t)$$

is the ReLU activation.

As we have seen in Section 6, we have $f^{(k)} = O(1)$ by choosing $\mathbf{w}_r \sim O(m^{-1/2k})$, therefore we restrict attention on such $\mathbf{W}$'s by considering the constraint set $\{\mathbf{W} : \|\mathbf{W}\|_{2,2k}^{2k} \leq B_w^{2k}\}$ for some $B_w = O_m(1)$.

**Overview of results** This subsection establishes the following results for the $k$-th order NTK.

- We bound the generalization of $f^{(k)}$ through the tensor operator norm of a certain $k$-tensor involving the features (Lemma 17). Consequently, the generalization of the $k$-th order NTK for $\|\mathbf{W}\|_{2,2k} \leq B_w$, when the base distribution of $\mathbf{x}$ is uniform on the sphere, scales as

$$\widetilde{O}\left(B_x^k B_w^k \left[\frac{1}{\sqrt{nd^{k-1}}} + \frac{1}{n}\right] + \frac{1}{\sqrt{n}}\right).$$

(Theorem 19). Compared with the distribution-free bound $B_x^k B_w^k / \sqrt{n}$, the leading term is better by a factor of $\sqrt{\min\{d^{k-1}, n\}}$. In particular, when $n \geq d^{k-1}$, the generalization is better by a factor of $\sqrt{d^{k-1}}$ than the distribution-free bound.

- For the polynomial $f_\star(\mathbf{x}) = \alpha(\boldsymbol{\beta}^\top \mathbf{x})^p$ with $p \geq k$ (and $p - k$ is even or one), when $m$ is sufficiently large, there exists a $\mathbf{W}_\star$ expressing $f_\star$ such that

$$\|\mathbf{W}_\star\|_{2,2k}^{2k} \leq O\left(p^3 \alpha^2 B_x^{2(p-k)} \|\boldsymbol{\beta}\|_2^{2p}\right).$$

(Theorem 20). Substituting into the generalization bound yields the following generalization error for learning $f_\star$:

$$\widetilde{O}\left(p^3 \alpha^2 (B_x \|\boldsymbol{\beta}\|_2)^p \left[\frac{1}{\sqrt{nd^{k-1}}} + \frac{1}{n}\right]\right).$$

In particular, the leading multiplicative factor is the same for all $k$ (including the linear NTK with $k = 1$), but the sample complexity is lower by a factor of $d^{k-1}$ when $n \geq d^{k-1}$. This shows systematically the benefit of higher-order NTKs when distributional assumptions are present.

**Tensor operator and nuclear norm** Our result requires the definition of operator norm and nuclear norm for $k$-tensors, which we briefly review here. The operator norm of a symmetric $k$-tensor $\mathbf{A} \in \mathbb{R}^{d^k}$ is defined as

$$\|\mathbf{A}\|_{\mathrm{op}} := \sup_{\|\mathbf{v}\|_2=1} \left\langle \mathbf{A}, \mathbf{v}^{\otimes k}\right\rangle = \sup_{\|\mathbf{v}\|_2=1} \mathbf{A}[\mathbf{v}, \ldots, \mathbf{v}].$$

The nuclear norm $\|\cdot\|_*$ is defined as the dual norm of the operator norm:

$$\|\mathbf{A}\|_* := \sup_{\|\mathbf{B}\|_{\mathrm{op}}=1} \langle \mathbf{A}, \mathbf{B}\rangle.$$

Specifically, for any rank-one tensor $\boldsymbol{u}^{\otimes k}$, we have

$$\left\|\boldsymbol{u}^{\otimes k}\right\|_* = \sup_{\|\mathbf{B}\|_{\mathrm{op}}=1} \left\langle \boldsymbol{u}^{\otimes k}, \mathbf{B}\right\rangle = \|\boldsymbol{u}\|_2^k,$$

i.e. its nuclear norm equals its operator norm (and also the Frobenius norm).

### D.2.1 GENERALIZATION

We begin by stating a generalization bound for $f^{(k)}$, which depends on the operator norm of a $k$-th order tensor feature, generalizing Lemma 5.

**Lemma 17** (Bounding generalization of $f^{(k)}$ via tensor operator norm). *For any non-negative loss $\ell$ such that $z \mapsto \ell(y, z)$ is 1-Lipschitz and $\ell(y, 0) \leq 1$ for all $y \in \mathcal{Y}$, we have the Rademacher complexity bound*

$$\mathbb{E}_{\boldsymbol{\sigma}, \mathbf{x}}\left[\sup_{\|\mathbf{W}\|_{2,2k} \leq B_w} \frac{1}{n}\sum_{i=1}^n \sigma_i \ell(y_i, f_{\mathbf{W}_0,\mathbf{W}}^{(k)}(\mathbf{x}_i))\right] \leq 2B_w^k \mathbb{E}_{\boldsymbol{\sigma},\mathbf{x}}\left[\max_{r \in [m]} \left\|\frac{1}{n}\sum_{i=1}^n \sigma_i \sigma_k(\mathbf{w}_{0,r}^\top \mathbf{x}_i) \mathbf{x}_i^{\otimes k}\right\|_{\mathrm{op}}\right] + \frac{1}{\sqrt{n}},$$

*where $\sigma_i \overset{\mathrm{iid}}{\sim} \mathrm{Unif}\{\pm 1\}$ are Rademacher variables.*

*Proof.* The proof is analogous to that of Lemma 5. As the loss $\ell(y, z)$ is 1-Lipschitz in $z$ for all $y$, by the Rademacher contraction theorem (Wainwright, 2019, Chapter 5) we have that

$$
\mathbb{E}_{\boldsymbol{\sigma}, \mathbf{x}}\left[\sup_{\|\mathbf{W}\|_{2,2k} \leq B_w} \frac{1}{n} \sum_{i=1}^{n} \sigma_i \ell(y_i, f_{\mathbf{W}_0, \mathbf{W}}^{(k)}(\mathbf{x}_i))\right]
$$

$$
\leq 2\mathbb{E}_{\boldsymbol{\sigma}, \mathbf{x}}\left[\sup_{\|\mathbf{W}\|_{2,2k} \leq B_w} \frac{1}{n} \sum_{i=1}^{n} \sigma_i f_{\mathbf{W}_0, \mathbf{W}}^{(k)}(\mathbf{x}_i)\right] + \mathbb{E}_{\boldsymbol{\sigma}, \mathbf{x}}\left[\frac{1}{n} \sum_{i=1}^{n} \sigma_i \ell(y_i, 0)\right]
$$

$$
\leq 2\mathbb{E}_{\boldsymbol{\sigma}, \mathbf{x}}\left[\sup_{\|\mathbf{W}\|_{2,2k} \leq B_w} \frac{1}{\sqrt{m}} \sum_{r \leq m} \left\langle \frac{1}{n} \sum_{i=1}^{n} \sigma_i a_r \sigma_k(\mathbf{w}_{0,r}^\top \mathbf{x}_i) \mathbf{x}_i^{\otimes k}, \mathbf{w}_r^{\otimes k} \right\rangle\right] + \frac{1}{\sqrt{n}}
$$

$$
\leq 2\mathbb{E}_{\boldsymbol{\sigma}, \mathbf{x}}\left[\sup_{\|\mathbf{W}\|_{2,2k} \leq B_w} \max_{r \in [m]} \left\|\frac{1}{n} \sum_{i=1}^{n} a_r \sigma_i \sigma_k(\mathbf{w}_{0,r}^\top \mathbf{x}_i) \mathbf{x}_i^{\otimes k}\right\|_{\mathrm{op}} \cdot \frac{1}{\sqrt{m}} \sum_{r \leq m} \|\mathbf{w}_r^{\otimes k}\|_*\right] + \frac{1}{\sqrt{n}}
$$

$$
\leq 2\mathbb{E}_{\boldsymbol{\sigma}, \mathbf{x}}\left[\max_{r \in [m]} \left\|\frac{1}{n} \sum_{i=1}^{n} \sigma_i \sigma_k(\mathbf{w}_{0,r}^\top \mathbf{x}_i) \mathbf{x}_i^{\otimes k}\right\|_{\mathrm{op}}\right] \cdot \underbrace{\sup_{\|\mathbf{W}\|_{2,2k} \leq B_w} \frac{1}{\sqrt{m}} \|\mathbf{w}_r\|_2^k}_{\leq B_w^k} + \frac{1}{\sqrt{n}},
$$

where the last step used the power mean (or Cauchy-Schwarz) inequality on $\{\|\mathbf{w}_r\|_2\}$. $\square$ $\square$

**Bound on tensor operator norm** It is straightforward to see that the expected tensor operator norm can be bounded as

$$
\widetilde{O}\big(B_x^k / \sqrt{n}\big)
$$

without any distributional assumptions on $\mathbf{x}$. We now provide a bound on the expected tensor operator norm appearing in Lemma 17 in the special case of uniform features, i.e. $\mathbf{x} \sim \mathrm{Unif}(\mathbb{S}^{d-1}(B_x))$.

**Lemma 18** (Tensor operator norm bound for uniform features). *Suppose $\mathbf{x}_i \overset{\mathrm{iid}}{\sim} \mathrm{Unif}(\mathbb{S}^{d-1}(B_x))$. Then for any $k \geq 3$ and $k = O(1)$, we have (with high probability over $\mathbf{W}_0$)*

$$
\mathbb{E}_{\boldsymbol{\sigma}, \mathbf{x}}\left[\max_{r \in [m]} \left\|\frac{1}{n} \sum_{i=1}^{n} \sigma_i \sigma_k(\mathbf{w}_{0,r}^\top \mathbf{x}_i) \mathbf{x}_i^{\otimes k}\right\|_{\mathrm{op}}\right] \leq \widetilde{O}\left(B_x^k \left[\frac{1}{\sqrt{nd^{k-1}}} + \frac{1}{n}\right]\right). \tag{31}
$$

Substituting the above bound into Lemma 17 directly leads to the following generalization bound for $f^{(k)}$:

**Theorem 19** (Generalization for $f^{(k)}$ with uniform features). *Suppose $\mathbf{x}_i \overset{\mathrm{iid}}{\sim} \mathrm{Unif}(\mathbb{S}^{d-1}(B_x))$. Then for any $k \geq 3$ and $k = O(1)$, we have (with high probability over $\mathbf{W}_0$)*

$$
\mathbb{E}_{\mathcal{D}}\left[\sup_{\|\mathbf{W}\|_{2,2k} \leq B_w} \left(L_P(f_{\mathbf{W}_0, \mathbf{W}}^{(k)}) - L(f_{\mathbf{W}_0, \mathbf{W}}^{(k)})\right)\right] \leq \widetilde{O}\left(B_x^k B_w^k \left[\frac{1}{\sqrt{nd^{k-1}}} + \frac{1}{n}\right] + \frac{1}{\sqrt{n}}\right).
$$

The proof of Lemma 18 is deferred to Appendix D.3.

### D.2.2 EXPRESSIVITY

**Theorem 20** (Expressivity of $f^{(k)}$). *Suppose $\{(a_r, \mathbf{w}_{0,r})\}$ are generated according to the symmetric initialization (3), and $f_\star(\mathbf{x}) = \alpha(\boldsymbol{\beta}^\top \mathbf{x})^p$ where $p - k \in \{1\} \cup \{2\ell\}_{\ell \geq 0}$. Suppose further that $\sigma$ is such that $\sigma_k(t) = \mathrm{relu}(t)$, then so long as the width is sufficiently large:*

$$
m \geq \widetilde{O}\big(ndp^3 \alpha^2 (B_x \|\boldsymbol{\beta}\|_2)^{2p} \epsilon^{-2}\big),
$$

*we have with probability at least $1 - \delta$ (over $\mathbf{W}_0$) that there exists $\mathbf{W}_\star \in \mathbb{R}^{d \times m}$ such that*

$$
\left|L^Q(\mathbf{W}_\star) - L(f_\star)\right| \leq \epsilon \quad \text{and} \quad \|\mathbf{W}_\star\|_{2,2k}^{2k} \leq B_{w,\star}^{2k} = O\left(p^3 \alpha^2 B_x^{2(p-k)} \|\boldsymbol{\beta}\|_2^{2p} \delta^{-1}\right).
$$

The proof of Theorem 20 is deferred to Appendix D.4.

### D.3 PROOF OF LEMMA 18

We begin by observing for any symmetric tensor $\mathbf{A} \in \mathbb{R}^{d^k}$ that

$$\|\mathbf{A}\|_{\mathrm{op}} \leq \frac{1}{1-k\epsilon} \sup_{\mathbf{v} \in N(\epsilon)} \mathbf{A}[\mathbf{v}, \ldots, \mathbf{v}],$$

where $N(\epsilon)$ is an $\epsilon$-covering of unit sphere $\mathbb{S}^{d-1}(1)$. (The proof follows by bounding $\mathbf{A}[\boldsymbol{u}, \ldots, \boldsymbol{u}]$ by $\mathbf{A}[\mathbf{v}, \ldots, \mathbf{v}] + k\epsilon \|\mathbf{A}\|_{\mathrm{op}}$ through replacing $\boldsymbol{u}$ by $\mathbf{v}$ one at a time). Taking $\epsilon = 1/(2k)$, we have

$$\mathbb{P}_{\boldsymbol{\sigma},\mathbf{x}} \left( \max_{r \in [m]} \left\| \frac{1}{n} \sum_{i=1}^{n} \sigma_i \sigma_k(\mathbf{w}_{0,r}^{\top} \mathbf{x}_i) \mathbf{x}_i^{\otimes k} \right\|_{\mathrm{op}} \geq t \right)$$

$$\leq \mathbb{P}_{\boldsymbol{\sigma},\mathbf{x}} \left( \max_{r \in [m], \mathbf{v} \in N(1/(2k))} \frac{1}{n} \sum_{i=1}^{n} \sigma_i \sigma_k(\mathbf{w}_{0,r}^{\top} \mathbf{x}_i)(\mathbf{v}^{\top} \mathbf{x}_i)^k \geq t/2 \right).$$

We now perform a truncation argument to upper bound the above probability. Let $M > 0$ be a truncation level to be determined, we have by the Bernstein inequality that

$$\mathbb{P}_{\boldsymbol{\sigma},\mathbf{x}} \left( \max_{r \in [m]} \left\| \frac{1}{n} \sum_{i=1}^{n} \sigma_i \sigma_k(\mathbf{w}_{0,r}^{\top} \mathbf{x}_i) \mathbf{x}_i^{\otimes k} \right\|_{\mathrm{op}} \geq t \right)$$

$$\leq \mathbb{P}_{\boldsymbol{\sigma},\mathbf{x}} \left( \max_{r \in [m], \mathbf{v} \in N(1/(2k))} \frac{1}{n} \sum_{i=1}^{n} \sigma_i \sigma_k(\mathbf{w}_{0,r}^{\top} \mathbf{x}_i)(\mathbf{v}^{\top} \mathbf{x}_i)^k \mathbf{1}\left\{ \left| \sigma_k(\mathbf{w}_{0,r}^{\top} \mathbf{x}_i) \right| \leq M \right\} \geq t/2 \right)$$

$$+ \mathbb{P}_{\mathbf{x}} \left( \max_{r,i} \left| \sigma_k(\mathbf{w}_{0,r}^{\top} \mathbf{x}_i) \right| \geq M \right)$$

$$\leq \exp\left( -c \min\left\{ \frac{nt^2}{\widetilde{O}(1) \cdot B_x^{2k} d^{-k}}, \frac{nt}{M B_x^k} \right\} + d\log 6k + \log m \right) + \exp\left( -\frac{M^2}{2\widetilde{O}(1)} + \log mn \right),$$

where the $\widetilde{O}(1) \cdot B_x^{2k} d^{-k}$ comes from computing the variance of

$$Z_i := \sigma_i \sigma_k(\mathbf{w}_{0,r}^{\top} \mathbf{x}_i)(\mathbf{v}^{\top} \mathbf{x}_i)^k$$

using that $\mathbf{x}_i$ are uniform on the sphere (see, e.g. (Ghorbani et al., 2019b, Proof of Lemma 4)); $M B_x^k$ is the bound on the variable $Z_i$, and the $\widetilde{O}(1)$ comes from the fact that $\|\mathbf{w}_{0,r}\|_2 \leq \widetilde{O}(\sqrt{d} B_x^{-1})$ with high probability. Now, choosing

$$M = (nt/B_x^k)^{1/2},$$

the above bound reads

$$\exp\left( -c \min\left\{ \frac{nt^2}{\widetilde{O}(1) \cdot B_x^{2k} d^{-k}}, \left( \frac{nt}{B_x^k} \right)^{1/2} \right\} + \widetilde{O}(d) \right) + \exp\left( -\frac{nt/B_x^k}{2\widetilde{O}(1)} + \widetilde{O}(1) \right) := p_t.$$

It remains to bound $\int_{t=0}^{\infty} p_t$ to give an expectation bound on the desired tensor operator norm. This follows by adding up the following three bounds:

(1) For the main branch "$nt^2/\widetilde{O}(B_x^{2k} d^{-k})$" we have

$$\int_0^{\infty} \min\left\{ \exp\left( -\frac{nt^2}{\widetilde{O}(B_x^{2k} d^{-k})} + \widetilde{O}(d) \right), 1 \right\} dt \leq \widetilde{O}\left( \sqrt{\frac{B_x^{2k}}{nd^{k-1}}} \right).$$

This follows by integrating the "1" branch for $t \leq \widetilde{O}(\sqrt{B_x^{2k} d^{-(k-1)}/n})$ (which yields the right hand side) and integrating the other branch otherwise (the integral being upper bounded by $\widetilde{O}(\sqrt{B_x^{2k} d^{-k}/n})$, dominated by the right hand side).

(2) The branch "$(nt/B_x^k)^{1/2}$" is taken only when

$$\left( \frac{nt}{B_x^k} \right)^{1/2} < \frac{nt^2}{\widetilde{O}(B_x^{2k} d^{-k})} \quad \text{i.e.} \quad t > \widetilde{O}\left( n^{-1/3} B_x^k d^{2k/3} \right).$$

On the other hand, the inequality $(nt/B_x^k)^{1/2} > \widetilde{O}(d)$ happens when

$$t > \widetilde{O}\big(d^2 B_x^k/n\big),$$

which is implied by the preceding condition so long as $k \geq 3$. Therefore, when this branch is taken, the $\widetilde{O}(d)$ can already be absorbed into the main term, so the contribution of this branch can be bounded as

$$\int_{\widetilde{O}(n^{-1/3}B_x^k d^{2k/3})}^{\infty} \exp\left(-c'\left(\frac{nt}{B_x^k}\right)^{1/2}\right) dt \leq \int_0^{\infty} \exp\left(-c'\left(\frac{nt}{B_x^k}\right)^{1/2}\right) dt \leq O\left(\frac{B_x^k}{n}\right).$$

(3) We have

$$\int_0^{\infty} \min\left\{\exp\left(-\frac{nt/B_x^k}{2\widetilde{O}(1)} + \widetilde{O}(1)\right), 1\right\} dt \leq \widetilde{O}(B_x^k/n),$$

using a similar argument as part (1).

Putting together the above three bounds, we obtain

$$\max_{r \in [m]} \left\|\frac{1}{n}\sum_{i=1}^{n} \sigma_i \sigma_k(\mathbf{w}_{0,r}^\top \mathbf{x}_i)\mathbf{x}_i^{\otimes k}\right\|_{\mathrm{op}} \leq \int_0^{\infty} p_t dt \leq \widetilde{O}\left(B_x^k\left[\frac{1}{\sqrt{nd^{k-1}}} + \frac{1}{n}\right]\right),$$

the desired result. $\qquad\square$

## D.4 Proof of Theorem 20

Our proof is analogous to that of Theorem 16, in which we first look at the case of infinitely many neurons and then use concentration to carry the result onto finitely many neurons.

**Expressivity with infinitely many neurons**  We first consider expressing $f_\star(\mathbf{x}) = \alpha(\boldsymbol{\beta}^\top \mathbf{x})^p$ with infinite-neuron version of $f^{(k)}$, that is, we wish to find random variables $(\mathbf{w}_+, \mathbf{w}_-)$ such that

$$\mathbb{E}_{\mathbf{w}_0}\big[\mathrm{relu}(\mathbf{w}_0^\top \mathbf{x})\big((\mathbf{w}_+^\top \mathbf{x})^k - (\mathbf{w}_-^\top \mathbf{x})^k\big)\big] = f_\star(x).$$

Choosing

$$(\mathbf{w}_+, \mathbf{w}_-) = \left(([a]_+)^{1/k}\boldsymbol{\beta}, ([a]_-)^{1/k}\boldsymbol{\beta}\right)$$

for some real-valued random scalar $a$ (that depends on $\mathbf{w}_0$), we have

$$\mathbb{E}_{\mathbf{w}_0}\big[\mathrm{relu}(\mathbf{w}_0^\top \mathbf{x})\big((\mathbf{w}_+^\top \mathbf{x})^k - (\mathbf{w}_-^\top \mathbf{x})^k\big)\big] = (\boldsymbol{\beta}^\top \mathbf{x})^k \cdot \mathbb{E}_{\mathbf{w}_0}\big[\mathrm{relu}(\mathbf{w}_0^\top \mathbf{x})a\big],$$

therefore the task reduces to finding $a = a(\mathbf{w}_0)$ such that $\mathbb{E}_{\mathbf{w}_0}\big[\mathrm{relu}(\mathbf{w}_0^\top \mathbf{x})a\big] = \alpha(\boldsymbol{\beta}^\top \mathbf{x})^{p-k}$. By Lemma 9, there exists $a = a(\mathbf{w}_0)$ satisfying the above and such that

$$\mathbb{E}_{\mathbf{w}_0}[a^2] \leq 2\pi((p-k)\vee 1)^3 \alpha^2 B_x^{2(p-k)} \|\boldsymbol{\beta}\|_2^{2(p-k)}. \tag{32}$$

Using this $a$, the $k$-th order NTK defined by $(\mathbf{w}_+, \mathbf{w}_-)$ expresses $f_\star$ and further satisfies the bound

$$\mathbb{E}_{\mathbf{w}_0}\left[\|\mathbf{w}_+\|_2^{2k} + \|\mathbf{w}_-\|_2^{2k}\right] = \mathbb{E}_{\mathbf{w}_0}[a^2] \cdot \|\boldsymbol{\beta}\|_2^{2k} \leq 2\pi((p-k)\vee 1)^3 \alpha^2 B_x^{2(p-k)} \|\boldsymbol{\beta}\|_2^{2p}.$$

**Finite neurons**  Given the symmetric initialization $\{\mathbf{w}_{0,r}\}_{r=1}^m$, for all $r \in [m]$, we consider $\mathbf{W}_\star \in \mathbb{R}^{d \times m}$ defined through

$$(\mathbf{w}_{\star,r}, \mathbf{w}_{\star,r+m}) = \left(m^{-1/2k}\mathbf{w}_+(\mathbf{w}_{0,r}), m^{-1/2k}\mathbf{w}_-(\mathbf{w}_{0,r})\right),$$

where we recall $(\mathbf{w}_+(\mathbf{w}_0), \mathbf{w}_-(\mathbf{w}_0)) = (a_+(\mathbf{w}_0)^{1/k}\boldsymbol{\beta}, a_-(\mathbf{w}_0)^{1/k}\boldsymbol{\beta})$. We then have

$$f_{\mathbf{W}_0, \mathbf{W}_\star}^{(k)}(\mathbf{x}) = \frac{1}{\sqrt{m}}\sum_{r \leq m} \sigma_k(\mathbf{w}_{0,r}^\top \mathbf{x})\big[(\mathbf{w}_{\star,r}^\top \mathbf{x})^k - (\mathbf{w}_{\star,r+m}^\top \mathbf{x})^k\big]$$

$$= \frac{1}{m}\sum_{r \leq m} \sigma_k(\mathbf{w}_{0,r}^\top \mathbf{x})\big[(\mathbf{w}_+(\mathbf{w}_{0,r})^\top \mathbf{x})^k - (\mathbf{w}_-(\mathbf{w}_{0,r})^\top \mathbf{x})^k\big]$$

$$= \left[\frac{1}{m}\sum_{r \leq m} \sigma_k(\mathbf{w}_{0,r}^\top \mathbf{x})a(\mathbf{w}_{0,r})\right] \cdot (\boldsymbol{\beta}^\top \mathbf{x})^k.$$

**Bound on $\|\mathbf{W}_\star\|_{2,2k}$**   As $f_\star(\mathbf{x}) = \alpha(\boldsymbol{\beta}^\top \mathbf{x})^p$, (32) guarantees that the coefficient $a(\mathbf{w}_0)$ involved above satisfies that

$$R_a^2 := \mathbb{E}_{\mathbf{w}_0}[a(\mathbf{w}_0)^2] \leq 2\pi((p-k) \vee 1)^3 \alpha^2 B_x^{2(p-k)} \|\boldsymbol{\beta}\|_2^{2(p-k)} .$$

By Markov inequality, we have with probability at least $1 - \delta/2$ that

$$\frac{1}{m} \sum_{r \leq m} a(\mathbf{w}_{0,r})^2 \leq 4\pi((p-k) \vee 1)^3 \alpha^2 B_x^{2(p-k)} \|\boldsymbol{\beta}\|_2^{2(p-k)} \delta^{-1},$$

which yields the bound

$$\|\mathbf{W}\|_{2,2k}^{wk} = \sum_{r \leq 2m} \|\mathbf{w}_{\star,r}\|_2^{2k}$$

$$\leq \|\boldsymbol{\beta}\|_2^{2k} \cdot \sum_{r \leq m} m^{-1} a(\mathbf{w}_{0,r})^2 \leq 4\pi[(p-k)^3 \vee 1]\alpha^2 B_x^{2(p-k)} \|\boldsymbol{\beta}\|_2^{2p} \delta^{-1}.$$

**Concentration of function**   Let $f_m(\mathbf{x}) = \frac{1}{m} \sum_{r \leq m} \sigma_k(\mathbf{w}_{0,r}^\top \mathbf{x}) a(\mathbf{w}_{0,r})$. We now show the concentration of $f_m$ to $f_{\star,p-k}(\mathbf{x}) := \alpha(\beta^\top \mathbf{x})^{p-k}$ over the dataset $\{\mathbf{x}_1, \ldots, \mathbf{x}_n\}$. We perform a truncation argument: let $R$ be a large radius (to be chosen) satisfying

$$\mathbb{P}_{\mathbf{W}_0}\left( \sup_{r \in [m]} \|\mathbf{w}_{0,r}\|_2 \geq R B_x^{-1} \right) \geq 1 - \delta/2. \tag{33}$$

On this event we have

$$f_m(\mathbf{x}) = \frac{1}{m} \sum_{r \leq m} \sigma_k(\mathbf{w}_{0,r}^\top \mathbf{x}) a(\mathbf{w}_{0,r}) \mathbf{1}\left\{ \|\mathbf{w}_{0,r}\|_2 \leq R B_x^{-1} \right\} := f_m^R(\mathbf{x}).$$

Letting $f_{\star,p-k}^R(\mathbf{x}) := \mathbb{E}_{\mathbf{w}_0}[\sigma_k(\mathbf{w}_0^\top \mathbf{x}) a(\mathbf{w}_0) \mathbf{1}\{\|\mathbf{w}_0\|_2 \leq R B_x^{-1}\}]$, we have

$$\mathbb{E}_{\mathbf{W}_0}\left[ \left( f_m(\mathbf{x}) - f_{\star,p-k}^R(\mathbf{x}) \right)^2 \right] = \frac{1}{m} \mathbb{E}_{\mathbf{w}_0}\left[ \sigma_k(\mathbf{w}_0^\top \mathbf{x}) a^2(\mathbf{w}_0) \mathbf{1}\{\|\mathbf{w}_0\|_2 \leq R\} \right] \leq C \frac{R^2 R_a^2}{m}.$$

Applying Chebyshev inequality and a union bound, we get

$$\mathbb{P}\left( \max_i |f_m(\mathbf{x}_i) - f_{\star,p-k}(\mathbf{x}_i)| \geq t \right) \leq C \frac{n R^2 R_a^2}{m t^2}.$$

For any $\epsilon > 0$, by substituting in $t = \epsilon B_x^{-k} \|\boldsymbol{\beta}\|_2^{-k} /2$, we see that

$$m \geq O\left( n R^2 R_a^2 B_x^{2k} \|\boldsymbol{\beta}\|_2^{2k} \epsilon^{-2} \right) = O\left( n R^2 (p-k)^3 \alpha^2 B_x^{2p} \|\boldsymbol{\beta}\|_2^{2p} \epsilon^{-2} \right) \tag{34}$$

ensures that

$$\max_{i \in [n]} |f_m(\mathbf{x}_i) - f_{\star,p-2}^R(\mathbf{x}_i)| \leq \epsilon B_x^{-k} \|\boldsymbol{\beta}\|^{-k} /2. \tag{35}$$

Next, for any $\mathbf{x}$ we have the bound

$$\left| f_{\star,p-k}^R(\mathbf{x}) - f_{\star,p-k}(\mathbf{x}) \right| = \left| \mathbb{E}_{\mathbf{w}_0}[\sigma_k(\mathbf{w}_0^\top \mathbf{x}) a(\mathbf{w}_0) \mathbf{1}\{\|\mathbf{w}_0\|_2 > R\}] \right|$$

$$\leq \mathbb{E}[a(\mathbf{w}_0)^2]^{1/2} \cdot \mathbb{E}[\sigma_k(\mathbf{w}_0^\top \mathbf{x})^4]^{1/4} \cdot \mathbb{P}(\|\mathbf{w}_0\|_2 > R)^{1/4}$$

$$\leq R_a \cdot C/\sqrt{d} \cdot \mathbb{P}(\|\mathbf{w}_0\|_2 > R)^{1/4}.$$

Choosing $R$ such that

$$\mathbb{P}(\|\mathbf{w}_0\|_2 > R) \leq c \frac{\sqrt{d} \epsilon^4}{R_a B_x^{4k} \|\boldsymbol{\beta}\|_2^{4k}} \tag{36}$$

ensures that

$$\max_i \left| f_{\star,p-2}^R(\mathbf{x}_i) - f_{\star,p-2}(\mathbf{x}_i) \right| \leq \frac{\epsilon B_x^{-k} \|\boldsymbol{\beta}\|_2^{-k}}{2}. \tag{37}$$

Combining (35) and (37), we see that with probability at least $1 - \delta$,

$$\max_{i \in [n]} \left| f^Q_{\mathbf{W}_\star}(\mathbf{x}_i) - f_\star(\mathbf{x}_i) \right| = \max_{i \in [n]} |f_m(\mathbf{x}_i) - f_{\star,p-k}(\mathbf{x}_i)| \cdot (\boldsymbol{\beta}^\top \mathbf{x}_i)^k$$

$$\leq 2 \cdot \frac{\epsilon B_x^{-k} \|\boldsymbol{\beta}\|_2^{-k}}{2} \cdot B_x^k \|\boldsymbol{\beta}\|_2^k = \epsilon$$

and thus

$$\left| L^Q(\mathbf{W}_\star) - L(f_\star) \right| \leq \epsilon. \tag{38}$$

To satisfy the requirements for $m$ and $R$ in (36) and (34), we first set $R = \widetilde{O}(\sqrt{d})$ (with sufficiently large log factor) to satisfy (36) by standard Gaussian norm concentration (cf. Appendix A.3), and by (34) it suffices to set $m$ as

$$m \geq \widetilde{O}\big(nd[(p-k)^3 \vee 1]\alpha^2(B_x \|\boldsymbol{\beta}\|_2)^{2p}\epsilon^{-2}\big).$$

for (38) to hold. $\qquad\square$

