# OpenReview forum: "Beyond Linearization: On Quadratic and Higher-Order Approximation of Wide Neural Networks"
_ICLR.cc/2020/Conference — Accept (Poster)_

### Official Review · AnonReviewer1 · 2019-10-23
**Official Blind Review #1**

**Rating:** 6

**Review:**

This paper studies the training of over-parameterized neural networks. Specifically, the authors propose a novel method to study the training beyond the neural tangent kernel regime by randomizing the network and eliminating the effect of the first order term in the network’s Taylor expansion. Both optimization guarantee and generalization error bounds are established for the proposed method. It is also shown that when learning polynomials, the proposed randomized networks outperforms NTK by a factor of d, where d is the input dimension.

Overall, I enjoy reading this paper. The presentation is clear, the arguments are insightful, and the proofs seem to be solid. Moreover, this paper offers some interesting ideas to show that neural networks can outperform NTK, which might be impactful. However, this paper also has certain weak points, mainly due to the less common problem setting.

Although it is believed that NTK cannot fully explain the success of deep learning, results in the NTK regime have the advantage that the problem setting (initialization method, training algorithm) is very close to what people do in practice. Therefore, ideally, a result beyond NTK should demonstrate the advantage of NN over NTK under similar, or at least practical settings. If the problem setting is changed in some strange way, then it might not be that meaningful even if the training behavior is different from the NTK setting. In my opinion, the following four points about the problem setting in this paper are less desired:

(1) Assumption A is not satisfied by any commonly used activation functions. The authors only provided cubic ReLU as an example.

(2) The randomization technique in this paper is not standard, and is pretty much an artifact to make the first order term in the Taylor expansion of neural networks small.

(3) The $(\| \cdot \|_{2,4})^8$ regularization term introduced on page 6 is highly unusual. Due to reparameterization, this regularization is on the distance towards initialization, indead of the norms of the weight parameters.

(4) The training algorithm used in this paper is noisy SGD due to the need to escape from saddle points.

Despite the issues mentioned above, I still think this paper is of good quality, and these limitations are understandable considering the difficulty to escape from the NTK regime. It would be interesting if the authors could provide some direct comparison between the generalization bound in this submission and existing generalization bounds in the NTK regime, for example the results in

Yuan Cao, Quanquan Gu, Generalization Error Bounds of Gradient Descent for Learning Over-parameterized Deep ReLU Networks
Yuan Cao, Quanquan Gu, Generalization Bounds of Stochastic Gradient Descent for Wide and Deep Neural Networks

Moreover, since this paper studies optimization with regularization on the distance towards initialization, it would also be nice to compare with the following paper:

Wei Hu, Zhiyuan Li, Dingli Yu, Understanding Generalization of Deep Neural Networks Trained with Noisy Labels


**Experience Assessment:**

I have published one or two papers in this area.

**Review Assessment: Checking Correctness Of Derivations And Theory:**

I carefully checked the derivations and theory.

**Review Assessment: Checking Correctness Of Experiments:**

N/A

**Review Assessment: Thoroughness In Paper Reading:**

I read the paper thoroughly.

---

> ### Author Response · Authors · 2019-11-08
> **Response**
>
> Thank you for your thoughtful and detailed comments. As you have raised, we also think that it’s important to understand how to provably escape the NTK regime for training wide neural nets, and we are glad that you liked our attempt.
>
> We address the specific concerns below:
>
> — Problem settings
> As pointed out, our setting is indeed different from (and slightly more non-standard than) the vanilla/NTK setting for learning wide two-layer neural nets. We want to highlight though that most of these settings are purely for technical reasons in order to analyze the quadratic model, which we clarify below:
> Re (1): The activation needs to be at least continuously twice differentiable in order for the second-order expansion to exist and for the loss to be $C^2$;
> Re (3): Noisy SGD is required to efficiently escape saddles;
> Re (4): The high-order regularization is also needed so that optimization never leaves a big $\|\cdot\|_{2,4}$ ball, so the coupling result holds.
> Tweaks like (3,4) are also present in e.g. Allen-Zhu et al. 2018 in order to provably learn a three-layer network.
>
> Re (2): This randomization step is the only key modification we’ve done (adding a $\{-1, 1\}$ multiplicative noise), which is the crucial step allowing us to escape the NTK and couple with a quadratic model.
> * On the one hand, this randomization is arguably quite similar to dropout, which essentially adds $\{0, 1\}$ multiplicative noise. We’d like to emphasize though that our randomization technique can be extended systematically—-beyond such dropout-like noise—-to obtain k-th order NTKs for all k (cf. Section 6).
> * On the other hand, we do think it is important to figure out how to train wide neural nets in quadratic / higher-order regimes without randomization, which will be an interesting direction for future work.
>
> — Comparison with (Cao & Gu, 2019ab) on generalization bounds & (Hu et al. 2019) on regularizer.
> We have cited these papers and commented on the comparison in our revision. Here we comment more on the comparison on the generalization bounds given in Section 5.2.
>
> To achieve a low generalization error, (Cao & Gu 2019ab) require that there exists a ground truth function with low RKHS norm in the random feature / NTK space that achieves low regression loss / large classification margin. In this same setting, our generalization bound (through learning a quadratic-like model) matches the above, but can in addition be better a dimension factor under mild distributional assumptions such as isotropic features. This happens since the quadratic model expresses high-degree polynomials in a more “compact” fashion than the linearized ones, which allows us to bound the generalization error through the *operator norm* of a certain matrix feature (cf. Theorem 6); in comparison, generalization bound obtained from linearized models would depend on the *Frobenius norm* of a similar feature matrix.

---

### Official Review · AnonReviewer3 · 2019-10-23
**Official Blind Review #3**

**Rating:** 6

**Review:**

This paper investigates higher order Taylor expansion of NTK (neural tangent kernel) beyond the linear term. The motivation of this study is to investigate the significant performance gap between NTK and deep neural network learning. The conventional NTK analysis only deals with the first order term of the Taylor expansion of the nonlinear activation, but this paper deals with higher order terms. In particular, it thoroughly investigates the second order expansion. It is theoretically shown that the expected risk can be approximated by the quadratic approximation, and show that the optimal solution under a quadratic approximation can achieve nice generalization error under some conditions.

Overall, I think the analysis interesting. Actually, there are big gap between NTK and real deep learning. However, this gap is not well clarified from theoretical view points. This is one of such attempts.
As far as I understand the analysis is solid. The writing is clear. I could easily understand the motivation and the results. The quadratic expansion is clearly different from the recent series of NTK analyses. In that sense, this study has sufficient novelty.

On the other hand, I think the study has several limitations. My concerns are summarized as follows:
- The proposed objective function is different from the normal objective function used for naive SGD because there is a "random initialization" term and some special regularization terms. Therefore, the analysis in this paper does not give precise understanding on what is going on for the naive SGD in deep learning.
- As far as I checked, there is no definition of OPT. Is it the optimal value of \tilde{L}(W)? Since OPT is an important quantity, this must be clarified.
- L^Q(W) considers essentially a quadratic model, and is different from the original model. It is unclear how expressive the quadratic model is. Since the region where the quadratic model is meaningful is restricted (i.e., ||w_r|| = O(m^{-1/4}) is imposed), the expressive power of the model in this regime is not obvious. It is nice if there are comments on how large its expressive power is. In particular, it is informative if sufficient conditions for L^Q(W_*) <= OPT is clarified.
- (This comment is related to the right above comment) There are two examples in which the linear NTK is outperformed by the quadratic model. However, they are rather simple. It would be better if there was more general (and practically useful) characterization so that there appears difference between the two regimes.

**Experience Assessment:**

I have read many papers in this area.

**Review Assessment: Checking Correctness Of Derivations And Theory:**

I assessed the sensibility of the derivations and theory.

**Review Assessment: Checking Correctness Of Experiments:**

N/A

**Review Assessment: Thoroughness In Paper Reading:**

I read the paper at least twice and used my best judgement in assessing the paper.

---

> ### Author Response · Authors · 2019-11-08
> **Response**
>
> Thank you for your thoughtful and detailed comments, and we are glad that you appreciate our motivation and liked our presentation. We address the specific concerns below:
>
> — Non-standard setting
> Our setting does contain a few tweaks that make it different from standard vanilla/NTK-regime for training a wide two-layer network. However, most of the tweaks (specifically the multiplicative randomization and the regularizer) are to help escape the NTK regime. Without such randomization, it is unclear/still open how to train wide neural nets beyond the NTK regime with provable guarantees, which we think is indeed an important future direction.
>
> — Definition of OPT
> OPT is defined throughout the statements of Lemma 1 - Theorem 4 as the optimal value of $L^Q(W)$ inside a 2, 4-norm ball.
>
> We have clarified our choice of OPT in the concrete examples later (e.g. in the sample complexity comparison in Section 5.2) in our revision.
>
> — Expressivity through $L^Q$, and conditions for OPT
> Due to an aggregation effect, $f^Q$ is able to express $O(1)$ functions even though each individual weight is only $O(m^{-1/4})$. We illustrated this in Section 3 in the comparison between $f^L$ and $f^Q$ [More precisely, when $\|w_r\|_2 = O(m^{-1/4})$, …]
>
> We have also updated our Theorem 7 in our revision to clarify conditions (as well as specify our choice of OPT) under which $L^Q(W_\star)\le {\rm OPT}$ is satisfied for a fairly broad class of functions (sum of “one-directional” polynomials.)
>
> — Comparison between quadratic model and NTK
> Our expressivity result (Theorem 7) works so long as there exists an underlying “sum of (one-directional) polynomial” type function that achieves low loss, and thus can be fairly general. This includes a single polynomial, or functions such as XOR that can be written as the sum of O(1) polynomials as is done in Section 5.2.
>
> To further demonstrate the generality of our sample complexity results, we have added an additional example of low-rank matrix sensing in our revision (Section 5.2), in which we show that the sample complexity upper bound for the quadratic model is also O(d) lower than that of the NTK.

---

### Official Review · AnonReviewer4 · 2019-10-26
**Official Blind Review #4**

**Rating:** 6

**Review:**

This paper presents an approach for going beyond NTK regime, namely the linear Taylor approximation for network function. By employing the idea of randomization this paper manages to reduce the magnitude of the linear part while the quadratic part is unaffected. This technique enables further analysis of both the optimization and generalization based on the nice property of quadratic approximation.

I believe this paper should be accepted because of its novel approach to go beyond the linear part, which I view as a main contribution. Also, this paper is well written and presents its optimization result and proof in a clear manner. Although it is an innovation in the theoretical perspective, I still want to raise two questions about the object this paper trying to analyze:

1. The activation function is designed so that it has really nice property: it has second-order derivative which is lipshitz. Actually I believe Assumption A is first motivated by cubic ReLU. Why is cubic ReLU so favorable in this paper? Is it possible to use quadratic ReLU in the proof? Also I wonder what is the key property of activation function which is allowed here.

2. The network model considered here, is modified to a randomized neural network. So maybe this paper just circumvents the difficulty in going beyond NTK regime? I have this concern because in reality optimizing this network model seems intractable. To evaluate the loss $L(\mathbf{W})$ or $L_{\lambda}(\mathbf{W})$ we take expectation over $\Sigma$; when doing gradient descent, apparently the gradient also takes exponential time to compute.

Overall, I believe this paper has high quality and I enjoyed reading it. However, I do expect response from authors which could address my concerns raised above. This can help me achieve a better understanding and a more precise evaluation on this paper.

***

I have read the author's response and decide to remain the rating as weak accept.

**Experience Assessment:**

I have read many papers in this area.

**Review Assessment: Checking Correctness Of Derivations And Theory:**

I assessed the sensibility of the derivations and theory.

**Review Assessment: Checking Correctness Of Experiments:**

N/A

**Review Assessment: Thoroughness In Paper Reading:**

I read the paper at least twice and used my best judgement in assessing the paper.

---

> ### Author Response · Authors · 2019-11-08
> **Response**
>
> Thank you for the thoughtful feedback, and we are glad that you liked our idea and presentation. We address the concerns below:
>
> — Assumption on activation function.
> The requirement on $\sigma$ in this paper comes from two sources:
> (1) Smoothness, so that the loss is twice *continuously* differentiable.
> (2) $\sigma’’$ is a sufficiently expressive nonlinearity, e.g. such that we can prove concrete results for $\sigma’’$ to express functions, e.g. polynomials (Theorem 7).
> A quadratic ReLU satisfies (2) ($\sigma’’$ is an indicator, which is at least as expressive as a ReLU) and *almost* (1), in that the loss will be twice differentiable (in a proper sense) but the second derivative will not be continuous, where in contrast a cubic ReLU has a continuous second derivative. We prefer to have this continuity, as existing convergence results on escaping-saddle algorithms require such continuity/Lipschitzness of the Hessian (see e.g. Jin et al. 2019). Apart from such an algorithmic concern, all our landscape results in Section 4 will hold if $\sigma$ is the quadratic ReLU (and we formally take $\sigma’’(t) = 1\{t \ge 0\}$.)
>
> — Computational cost of optimizing randomized network.
> As mentioned below Theorem 4, we use stochastic gradient descent to efficiently optimize the loss: at iteration $k$, we sample a fresh $\Sigma_k$ and perform a gradient step on $\widetilde{L}_\lambda(W\Sigma_k)$. As $L_\lambda(W)=E_\Sigma[\widetilde{L}_\lambda(W\Sigma)]$, the above procedure gives an unbiased estimate of $\nabla L_\lambda(W)$ which only requires polynomial time to compute. Further, with such additional stochasticity, the algorithm (SGD) will still converge to second-order stationary points in polynomial time, by existing results on escaping saddle points via SGD (Theorem 16, Jin et al. 2019). Such an algorithm is in fact very similar to adding dropout noise in practice, which uses i.i.d. $\{0,1\}$ multiplicative noise (instead of our $\{-1, 1\}$ noise) over the neurons at each iteration.

---

### Author Response · Authors · 2019-11-08
**Revision uploaded**

We have uploaded a revision of our paper to add in some updates and address the reviews. The main changes in this revision are below:

— Restated expressivity result with more generality
We have updated our expressivity result (Theorem 7) to cover the case where the ground truth is a *sum* of one-directional polynomials, rather than a single polynomial as we previously had. Such an extension was already (implicitly) used in Section 5.2 in our original version, and here we’d like to explicitly state it in this form for more clarity.

— Additional example on comparing quadratic model and NTK: low-rank matrix sensing
We have added an additional example of low-rank matrix sensing in our comparison between sample complexities in Section 5.2. Using our randomized quadratic-like net, the sample complexity upper bound for achieving $\epsilon$ test loss in this problem is $\widetilde{O}(d)$ better than that of linear NTK.

— Additional results and reorganization for higher-order NTKs
We have added generalization and expressivity results for k-th order NTKs for all $k\ge 2$ as a systematic extension of our results in the quadratic case. These results can be found in Appendix D.2. Due to space constraint, the original randomized coupling argument between neural nets and $k$-th order NTKs is now migrated to Appendix D.1.

Apart from the above changes, we have also added some additional citations and fixed certain typos.

---

### Public Comment · ~Greg_Yang1 · 2019-11-08
**Comparison with Neural Tangent Hierarchy**

Dear Authors,

Thanks for your interesting paper. How do the higher-order NTKs in your case compare with the neural tangent hierarchy [1]?

[1] https://arxiv.org/abs/1909.08156

---

> ### Author Response · Authors · 2020-02-15
> **Differences between Taylorized models (higher-order NTK) and NTH**
>
> Hi Greg,
>
> Thank you for the very insightful question! It is indeed useful to think about our higher-order NTK and the Neural Tangent Hierarchy (NTH) together, as both regimes refine the NTK theory.
>
> As far as we can tell now, there are two main differences between NTH and our higher-order NTK:
> -- NTH provides a *correction* to the NTK whereas higher-order NTK can potentially “escape” the NTK regime. More concretely, in NTH, each individual neuron only moves $O(1/\sqrt{m})$ (relative to its initial scale) similar as in the NTK regime. In contrast, when training is coupled with our quadratic model (e.g. our Theorem 4), each individual neuron moves $O(m^{-1/4})$ relatively, i.e. they move in a much larger ball than the NTK. This can have further consequences such as allowing provable convergence under a larger learning rate.
>
> -- NTH and our higher-order NTK refines the original NTK from different perspectives, and does not “cover” each other: 1) If we train a quadratic model and compute its NTH, then all the higher-order NTH terms would be non-vanishing, so that we need the actual infinite NTH hierarchy to accurately describe the training of a quadratic model; 2) Conversely, if we look at the NTH dynamics truncated at the first level higher than the NTK, then it does not correspond to training the Taylorized model (higher-order NTK) of any order.
>
> Despite these differences, one common thing about the NTH and our higher-order NTK is that they both provide finer approximations to the full training trajectory when training is actually in the NTK regime.

---

### Decision · Program_Chairs · 2019-12-19

**Decision:**

Accept (Poster)

**Comment:**

This paper studies the training of over-parameterized two-layer neural networks by considering high-order Taylor approximation, and randomizing the network to remove the first order term in the network’s Taylor expansion. This enables the neural network training go beyond the recently so-called neural tangent kernel (NTK) regime. The authors also established the optimization landscape, generalization error and expressive power results under the proposed analysis framework. They showed that when learning polynomials, the proposed randomized networks with quadratic Taylor approximation outperform standard NTK by a factor of the input dimension. This is a very nice work, and provides a new perspective on NTK and beyond. All reviewers are in support of accepting this paper.